# Synaptic and neural behaviours in a standard silicon transistor

Sebastian Pazos[1], Kaichen Zhu[1], Marco A. Villena[1], Osamah Alharbi[1], Wenwen Zheng[1], Yaqing Shen[1], Yue Yuan[1], Yue Ping[1] & Mario Lanza[2,3,4 ✉]

Hardware implementations of artificial neural networks (ANNs)—the most advanced of which are made of millions of electronic neurons interconnected by hundreds of millions of electronic synapses—have achieved higher energy efficiency than classical computers in some small-scale data-intensive computing tasks[1]. State-of-the-art neuromorphic computers, such as Intel's Loihi[2] or IBM's NorthPole[3], implement ANNs using bio-inspired neuron- and synapse-mimicking circuits made of complementary metal–oxide–semiconductor (CMOS) transistors, at least 18 per neuron and six per synapse. Simplifying the structure and size of these two building blocks would enable the construction of more sophisticated, larger and more energy-efficient ANNs. Here we show that a single CMOS transistor can exhibit neural and synaptic behaviours if biased in a specific (unconventional) manner. By connecting one additional CMOS transistor in series, we build a versatile 2-transistor-cell that exhibits adjustable neuro-synaptic response (which we named neuro-synaptic random access memory cell, or NS-RAM cell). This electronic performance comes with a yield of 100% and an ultra-low device-to-device variability, owing to the maturity of the silicon CMOS platform used—no materials or devices alien to the CMOS process are required. These results represent a short-term solution for the implementation of efficient ANNs and an opportunity in terms of CMOS circuit design and optimization for artificial intelligence applications.

Hardware-based ANNs are expected to outperform traditional computers in terms of energy efficiency because they can compute and store the data at the same location, which avoids energy consumption and delays related to data transfer. To do so, ANNs require (Fig. 1a): (1) electronic neurons capable of generating output signals that resemble a highly nonlinear hysteretic or thresholding mathematical operation when receiving several voltage or current excitatory inputs[4,5] and (2) electronic synapses capable of changing their electrical resistance to favour (facilitate) or limit (depress) the connections between specific neurons, a process that characterizes the learning of a feature. The persistence of these changes through time (plasticity) depends on the role of the synapse within the ANN and can be long or short term (see definitions in Supplementary Note 1 and Supplementary Fig. 1)[6]. In several types of ANNs, neural and synaptic behaviours take place dynamically (when applying electrical pulses rather than a continuous bias over time), which is highly desired for reducing energy consumption and providing synchronization with other parts of complex systems.

However, a single complementary metal–oxide–semiconductor (CMOS) transistor—as traditionally operated—cannot implement all these electrical behaviours, which has sparked a race to find the ideal hardware for implementing ANNs[7]. One solution is to implement electronic neurons and synapses using several interconnected CMOS transistors (Fig. 1b), but this approach implies large cost overheads that

arise from silicon area demands[8,9]. An alternative is to employ emerging device technologies, such as memristors (Fig. 1c)[10,11], but these solutions also face some implementation bottlenecks (peripherals and neural interfaces require dense CMOS circuits)[12,13], and they are still significantly smaller (with fewer neurons and synapses), less reliable and less widespread than purely CMOS implementations. Other innovative material platforms, such as ferroelectric, organic and two-dimensional materials[14–17], show promise but their integration challenges are far from being addressed.

In this scenario, it is logical to seek alternatives within existent technology platforms (Supplementary Table 1 and Supplementary Note 2). Two studies have proposed the implementation of electronic neurons and synapses based on home-made floating-gate silicon transistors[18,19], but the performance demonstrated was limited (especially for the synapse). More importantly, fabricating these devices is more complex and expensive than it is for CMOS transistors, and their integration density is lower because of the architecture and the nitridation thermal budgets[20]. One study implemented electronic neurons using partially depleted silicon-on-insulator tunnelling[21] field-effect transistors (FETs), but tunnelling FETs require specific doping profiles and gate alignment approaches. So far, the most technology-ready alternative for building electronic neurons is standard, partially depleted, silicon-on-insulator transistors operating in a band-to-band tunnelling regime[22–25] (used as

[1]Materials Science and Engineering, Physical Science and Engineering Division, King Abdullah University of Science and Technology (KAUST), Thuwal, Saudi Arabia. [2]Department of Materials Science and Engineering, National University of Singapore, Singapore, Singapore. [3]Institute for Functional Intelligent Materials, National University of Singapore, Singapore, Singapore. [4]Centre for Advanced 2D Materials (CA2DM), National University of Singapore, Singapore, Singapore. ✉e-mail: mlanza@nus.edu.sg

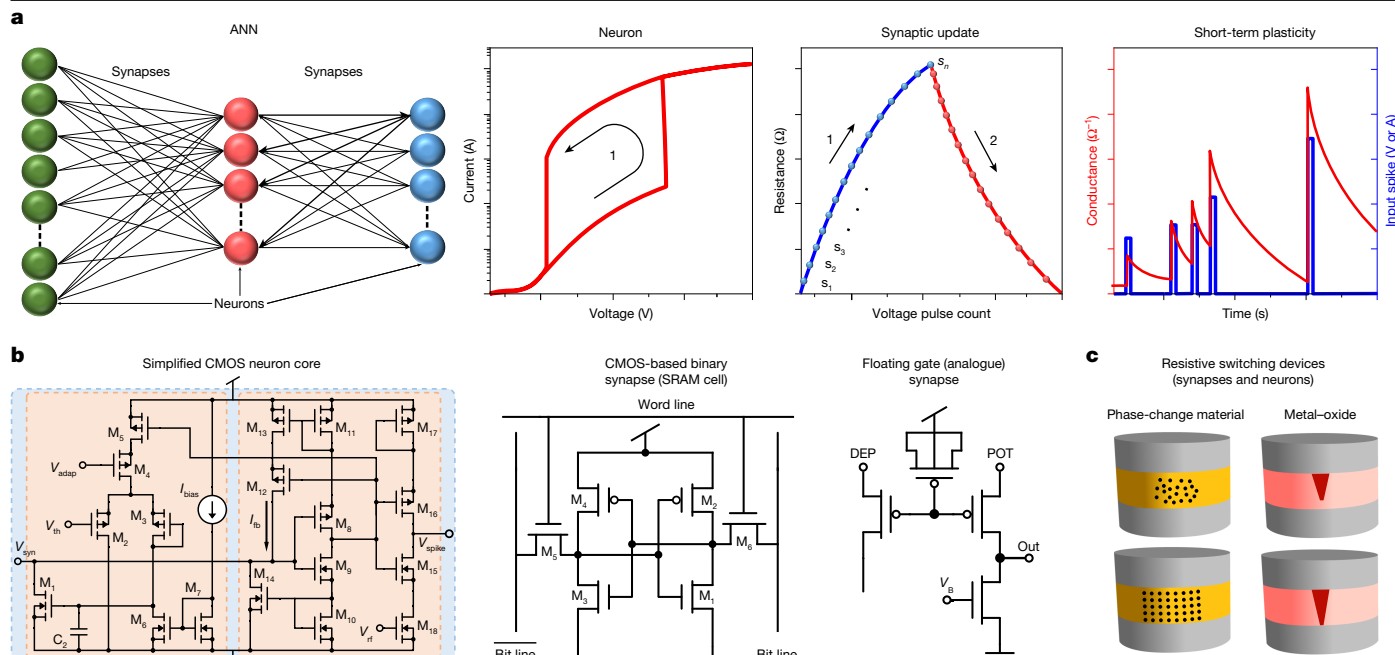

**Fig. 1 | Transistors and their use in neuro-synaptic-mimicking devices.** **a**, General structure of an ANN comprising neurons and synapses (left). Some of the typical $I$–$V$ or resistance behaviours of these devices are schematically represented: thresholding behaviour of neurons, update of the synaptic strength (weight) of synapses between neurons, and short-term plasticity displayed by synapses in spiking neural networks. **b**, Examples of electronic devices capable of mimicking some neuro-synaptic behaviours. Left, a single neuron core can comprise more than 20 CMOS transistors. Middle, the most basic digital cell used as a binary synapse is a six-transistor static random access memory cell. Right, an alternative implementation of a multilevel synapse using a floating-gate device. **c**, Representative examples of emerging memory devices, such as memristors based on phase-change materials or metals on oxides, which are being explored for their use as neurons and synapses in neuromorphic circuits. $s_n$, synaptic update input event (typically a voltage or current pulse); $V_{syn}$, post-synaptic signal as input to the neuron block; $I_{bias}$, neuron circuit reference bias current; $V_{th}$, threshold voltage setting; $V_{adap}$, input voltage for adaptive thresholding control; $I_{fb}$, feedback current; $V_{rf}$, output refractory period setting voltage; $V_{spike}$, spiking neuron output voltage; DEP, depression; POT, potentiation; $V_B$, bias voltage (read/update control); SRAM, static random access memory.

an integrator element in a leaky-integrate-and-fire neuron). However, to realize the thresholding operation, this approach still requires the interconnection of six devices. More importantly, demonstrations with partially depleted silicon-on-insulator transistors (both tunnelling and standard) only mimicked (limited) neuron functions, not synapses.

In this work, we experimentally demonstrate that excellent bio-inspired electronic neural and synaptic behaviours (Fig. 1a) can be mimicked in a single standard bulk-silicon metal–oxide–semiconductor field-effect transistor (MOSFET), if it is biased in a specific (unconventional) manner. More specifically, we operated the device on the verge of punch-through conditions while adjusting the resistance of the bulk connection to ground ($R_B$). When $R_B$ was implemented with another MOSFET, the resulting two-transistor cell could be continuously tuned between regimes to provide grand circuit-level versatility. In neuron mode, the two-transistor cell emulated leaky-integrate-and-fire neural behaviour and adaptive frequency bursting, with a high switching slope (below 10 mV dec$^{-1}$), large dynamic range (over 10$^3$), high endurance (over 10 million cycles) and competitive energy efficiency (firing energy down to 415 pJ μm$^{-1}$). When operated as a synapse, a single transistor in the floating-bulk configuration could be programmed at different (at least six) synaptic weights that were stable over time (long-term potentiation and depression) with high endurance (over 10$^5$ cycles). It was also capable of short-term pulsed facilitation, depression and synaptic plasticity with low variability and high robustness (up to 14 distinct levels for more than 700,000 potentiation and depression cycles).

## Punch-through impact ionization in MOSFETs

We evaluated the intrinsic hysteretic behaviour of an n-type MOSFET (channel length, $L_{CH}$ = 180 nm) operated in punch-through regime to

mimic neuron functionality. An equivalent circuit schematic including the parasitic devices for such a structure is displayed in Fig. 2a. When the bulk terminal was grounded, the typical output characteristic (drain current $I_D$ versus drain voltage $V_D$) of the MOSFET was detected, as expected (Fig. 2b (red lines) and Fig. 2c). However, when the bulk terminal was not directly grounded, the spreading resistance of the semiconductor substrate was large ($R_B$ in Fig. 2a), and drain-to-source voltage ($V_{DS}$) values in the range 2.5 to 3.5 V generated another $I_D$ component that resulted in abrupt current increases between 0.1 and 100 μA (Fig. 2b (blue lines) and Fig. 2d).

This phenomenon, which is highly dependent on the gate voltage $V_G$ (Supplementary Figs. 2 and 3), is related to the generation, through impact ionization, of excess electron–hole pairs in the vicinity of the drain and the deepening of the depletion region from the drain ($x_{pD}$) extending across the channel region (often referred to as punch-through; Fig. 2d). This regime was observed as early as 1987[26] and is characterized as the 'kink' effect in floating-body devices[27] (Methods). The physics of this phenomenon is succinctly reviewed in Supplementary Note 2. Driven by bulk currents (Methods and Supplementary Fig. 4), the process is largely dependent on the substrate spreading resistance and the effective resistance of the bulk connection. This resistance is different for each transistor in different chips and produces a very high variability in $I_D$ from one device to another (Supplementary Fig. 5). To solve this issue, we build a 2-transistor cell by including a bulk bias control MOSFET device (Fig. 2d) to tune the effective $R_B$ (between 10 kΩ and 1 MΩ; Fig. 2e) through the voltage applied to its gate, named $V_{G2}$, thus achieving total replicability of the process over 30 chips (Methods and Supplementary Figs. 6 and 7). This connection provides a collection path for excess carriers in the substrate and can be exploited to generate an abrupt thresholding

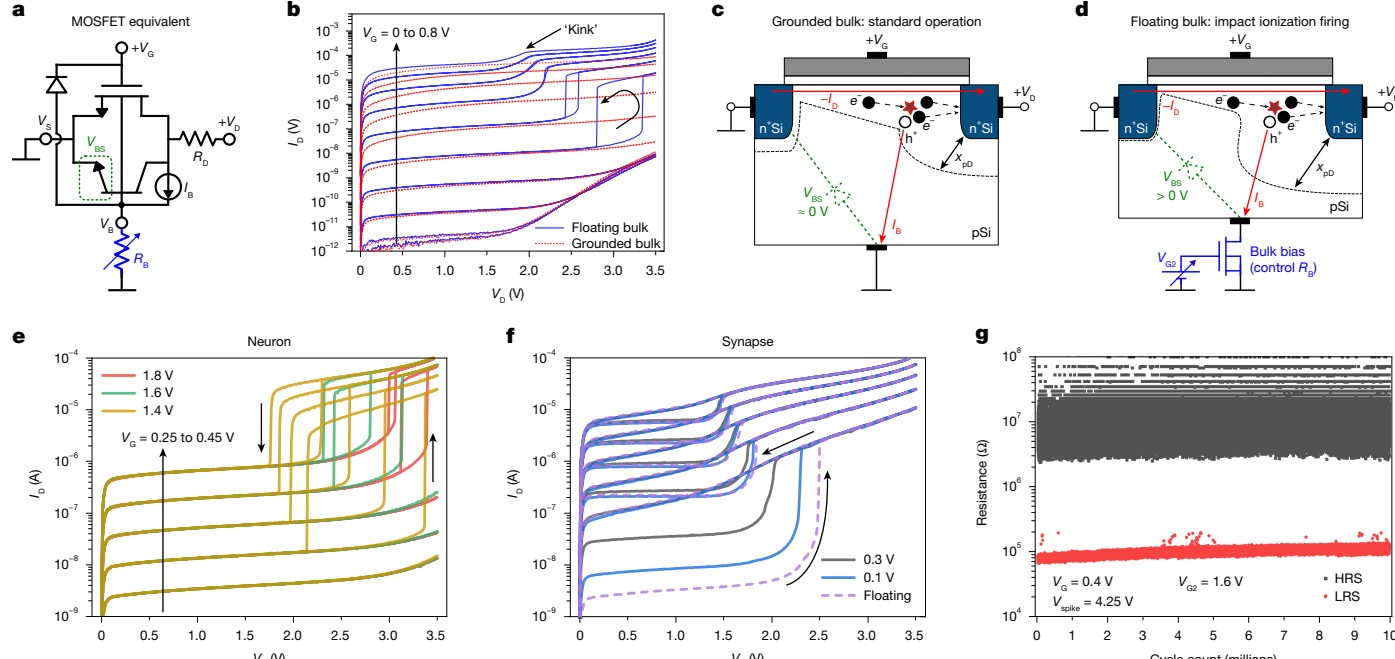

**Fig. 2 | Quasi-stationary I–V characteristics of a floating-bulk n-channel 180-nm MOSFET showing neuro-synaptic capabilities. a**, Simplified equivalent circuit of an n-channel MOSFET with parasitic substrate components. $I_B$ is the impact-ionization current component. $R_B$ and $R_D$ are the bulk and drain spreading resistances, respectively. $V_S$ is the source voltage. **b**, $I_D$–$V_D$ characteristics of an n-channel MOSFET. The normal conditions (grounded bulk, red curves) are compared with the floating-bulk condition (blue curves). **c**, Sketch of the current components in an n-channel MOSFET with a grounded-bulk connection and in the saturation condition (high $V_D$), where h$^+$ represents holes and $e^-$ electrons in the p-type silicon bulk. **d**, Same as **c** but under the floating-bulk condition,

where the bulk is floated through an external control device that regulates the bulk current ($I_B$) under punch-through avalanche conditions. **e,f**, $I_D$–$V_D$ characteristics of the MOSFET device depicted in **d** under different $V_{G2}$ (bulk bias control voltage) for a neuron (**e**) and a synapse (**f**). Different hysteretic behaviours are visible, from a typical thresholding neuron to the synaptic plasticity regime. In all cases, $V_G$ was varied between 0.25 and 0.45 V in 0.05 V steps. **g**, Robust and reproducible resistance change during thresholding of the floating-bulk device under fast ramp cycling for 10 million cycles (see Methods for measurement conditions). pSi, p-type silicon bulk; n$^+$Si, highly doped n-type silicon; $V_{BS}$, bulk-to-source voltage.

response, which is ideal for implementing integrate-and-fire neurons, or long-term conductance changes, adequate for implementing synapses (Fig. 2f). This extra transistor could be eliminated by engineering the bulk contact resistance for maximum integration density, although keeping it provides a circuit-level degree of freedom when tailoring the avalanche response to the application requirements or even dynamically within a single circuit.

We addressed the time-domain component of the firing and relaxation process that mimics a neuron by applying constant sweep rate $V_D$ ramps to the floating-bulk transistors and monitoring the value of $I_D$ (Methods). The results clearly show differences in the dependence of the firing and relaxing voltages on the sweep rate (Supplementary Figs. 8 and 9), which becomes shallower at higher $V_{G2}$. This is ascribed to a lower bulk resistance, which enables the collection of excess carriers in the bulk of the transistor and a quick drop of the electrostatic potential in the floating-bulk semiconductor at lower $R_B$. The physics behind these dynamics were carefully verified using technology computer-assisted design (TCAD) simulations replicating the experimental conditions (Methods) and showing excellent agreement with measurement results (see the detailed discussion on the simulation results in Supplementary Note 3).

## 2-transistor neuro-synaptic cell

We first investigated the potential of an n-type MOSFET operated in the punch-through impact-ionization regime for use as an electronic neuron in an ANN. For the application as a neuron, we needed the device to show, at first order, a thresholding (abrupt on/off behaviour) and hysteretic characteristic in the I–V curves, so we set the external bulk bias control voltage $V_{G2}$ in the 2-transistor cell to between 1.3 and 1.8 V to

a transistor with $L_{CH}$ = 180 nm (Fig. 2e). We observed that this hysteretic characteristic was extremely stable and repeatable through several iterations. We used high-speed ramps with peak value 4.2 V and with rates 80,000 V s$^{-1}$ and extracted the change in resistance across every cycle for 12 different CMOS transistors ($L_{CH}$ = 180 nm) for millions of cycles (Fig. 2g and Supplementary Fig. 15), which showed the robustness of the phenomenon. Similar behaviour was observed in CMOS transistors with $L_{CH}$ = 500 nm for more than 70,000 cycles (Supplementary Fig. 16). Under a pulsed firing regime (10-μs pulses and 60-μs relaxation transients), we also observed consistent behaviour over millions of cycles for $L_{CH}$ = 180 nm devices (Supplementary Fig. 17).

We next addressed the energy consumption of the device and obtained the firing time at constant applied drain voltage (Supplementary Fig. 18), depicted as $V_{spike}$ in reference to the spikes that a biological neuron integrates at its inputs (Methods). We measured firing times (Supplementary Figs. 19 and 20) from 10 μs to 2 ms for $V_{spike}$ between 3.5 and 4.5 V (Fig. 3a, left axis). The energy consumption of a single neuron was as low as 415 pJ μm$^{-1}$ (measured in devices with a channel width of 1 μm) at a firing time of 12.6 μs (Fig. 3a, right axis). These values are very competitive with other neuron-mimicking devices based on full-CMOS circuits[28] and CMOS integrated volatile memristors[29,30] (Supplementary Table 1). Our devices have the advantage of ease of integration and high tunability at little area overhead.

Similarly, we addressed the natural relaxation transients of the neural behaviour after firing a single pulse (which is known as the leaky characteristic of neurons; Supplementary Note 1). We performed time-resolved measurements at different $V_{G2}$ (0 to 1.8 V; Methods) and observed that the fired condition could be sustained over long periods of time exceeding several tens of milliseconds (Supplementary Fig. 21), resembling what is commonly known as synaptic plasticity[31]. We extracted the

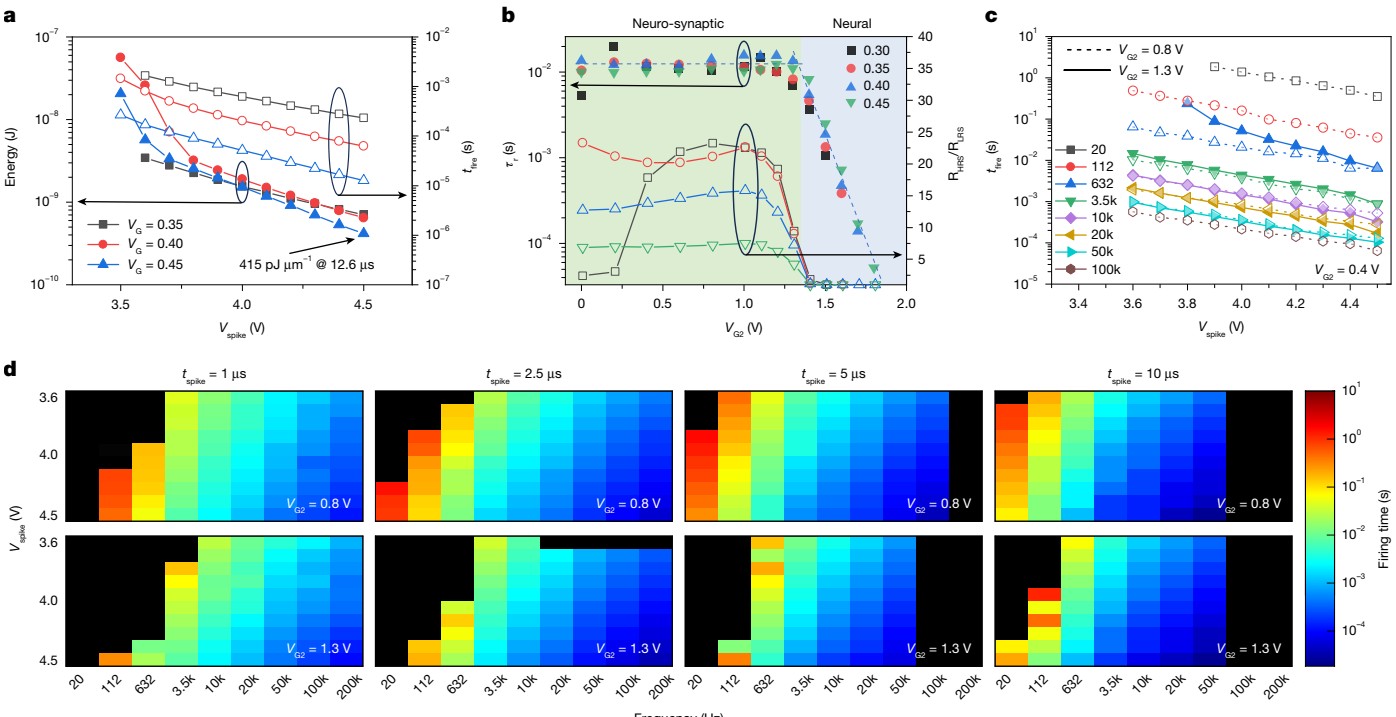

**Fig. 3 | Tuning the neural behaviour with punch-through control in floating-bulk 180-nm MOSFETs. a**, Mapping of the energy (left $y$ axis) and the firing time (right $y$ axis) as functions of $V_G$ and $V_{spike}$. **b**, Working regimes from neural to neuro-synaptic (right to left) tuned with the control voltage $V_{G2}$ and characterized by controllable $\tau_r$ (left $y$ axis) and synaptic weight (resistance) ratio $R_{HRS}/R_{LRS}$ (right $y$ axis). **c**, Dependence of the firing time ($t_{fire}$) on the spiking voltage ($V_{spike}$) and the input spiking frequency from 20 Hz to 100 kHz under different $V_{G2}$, The firing is controlled over four orders of magnitude. **d**, Heat maps of the tuning capabilities of the firing time for the floating-bulk device at different input spiking frequencies (20 Hz to 200 kHz), amplitudes (3.60–4.50 V) and spike durations (1–10 μs). Low $V_{G2}$ provides enhanced response at lower frequencies at the cost of slower relaxation times. Black tiles in the maps represent either no firing observed within 10 s or incompatible frequency versus $t_{spike}$ pairs (frequency greater than or equal to $t_{spike}^{-1}$).

characteristic relaxation time ($\tau_r$) and the synaptic update ratio (Methods) after a 30-ms window. Our results clearly indicate progressive tuning between purely neural behaviour ($V_{G2} > 1.3$ V) and neuro-synaptic behaviour ($V_{G2} < 1.3$ V), with characteristic times insensitive to $V_G$ (Fig. 3b, left axis). On the other hand, $V_G$ did affect the overall synaptic update ratio (Fig. 3b, right axis) and allowed or inhibited the firing for a fixed spike amplitude and duration. The operational flexibility that characterizes this 2-transistor cell makes it a convenient building block which we call neuro-synaptic random access memory cell, or NS-RAM.

Tuning the dynamics of excess carrier generation (through $V_G$) and the return to equilibrium after firing (through $V_{G2}$) allowed us to control the firing rate of the neuron under different excitatory voltage pulse amplitudes and frequencies (Methods). We measured the elapsed time until the neuron fired for each input (Supplementary Fig. 22). We observed a clear dependence of the firing time on the input spiking frequency and voltage, which could be inhibited at low frequencies by tuning $V_{G2}$ (Fig. 3c). We mapped these characteristics onto a $V_{spike}$ versus frequency space for spike durations $t_{spike}$ between 1 and 10 μs (Fig. 3d). We can clearly observe that the response could be tailored according to the system needs and to the process being mimicked. This behaviour is fundamental for replicating biological neural processes at different scales, for example the tonotopic mapping of audio signals performed by cells in the human cochlea[29] (Methods), and could be further tailored through external capacitances that impact the relaxation dynamics of the bulk semiconductor after firing (see discussion in Supplementary Note 3).

## Short-term plasticity in a single transistor

We next tackled the potential of the floating-bulk n-type MOSFET to operate as a synapse. To mimic this behaviour, we operated individual

transistors ($L_{CH} = 500$ nm) with a bipolar voltage scheme in the fully floating-bulk condition (the bulk bias control transistor was not connected, and the bulk terminal was left unbiased). We investigated the pulse modulation of the synaptic weight at the microsecond timescale and exploited the excess carrier dynamics in the floating-bulk device to mimic short-term synaptic dynamics. We used positive voltage spikes at the drain for potentiation (synaptic weight increase and reduction in resistance) and negative voltage spikes for depression (synaptic weight decrease and increase in resistance). We applied trains of potentiation and depression pulses. We read the resistance after each pulse (Methods) and observed the typical potentiation and depression behaviour of the synaptic weight (Fig. 4a) with excellent repeatability over 200,000 cycles (approximately 7 million pulses; Supplementary Fig. 23). We tuned the voltages and timing (Fig. 4b) and achieved low cycle-to-cycle variability (effective number of levels $N_{eff} = 14$; for detailed statistics, see Supplementary Fig. 23) and excellent linearity with a tuning range of approximately ×4 (nonlinearity exponential factors were 0.06 for potentiation and 0.21 for depression, where values below 1 typically represent good linearity[32]).

The synaptic weight range could be tuned by modifying the gate voltage applied to the device over two orders of magnitude (between 200 kΩ and 20 MΩ; Fig. 4c), which allowed us to customize the convenient operating range of the device within a specific circuit implementation. The potentiation process could be rapidly reset by a single, larger depression pulse that brought the synapse back to its quiescent state (Supplementary Fig. 24), with a synaptic update range reaching approximately ×10 and excellent repeatability over 760,000 full cycles (Fig. 4d) (Methods). Finally, we analysed the short-term synaptic plasticity[6,31] (Supplementary Note 1). We applied a potentiation sequence (learning) and addressed the synaptic decay through time (forgetting) under a constant read voltage. We observed characteristic exponential decay

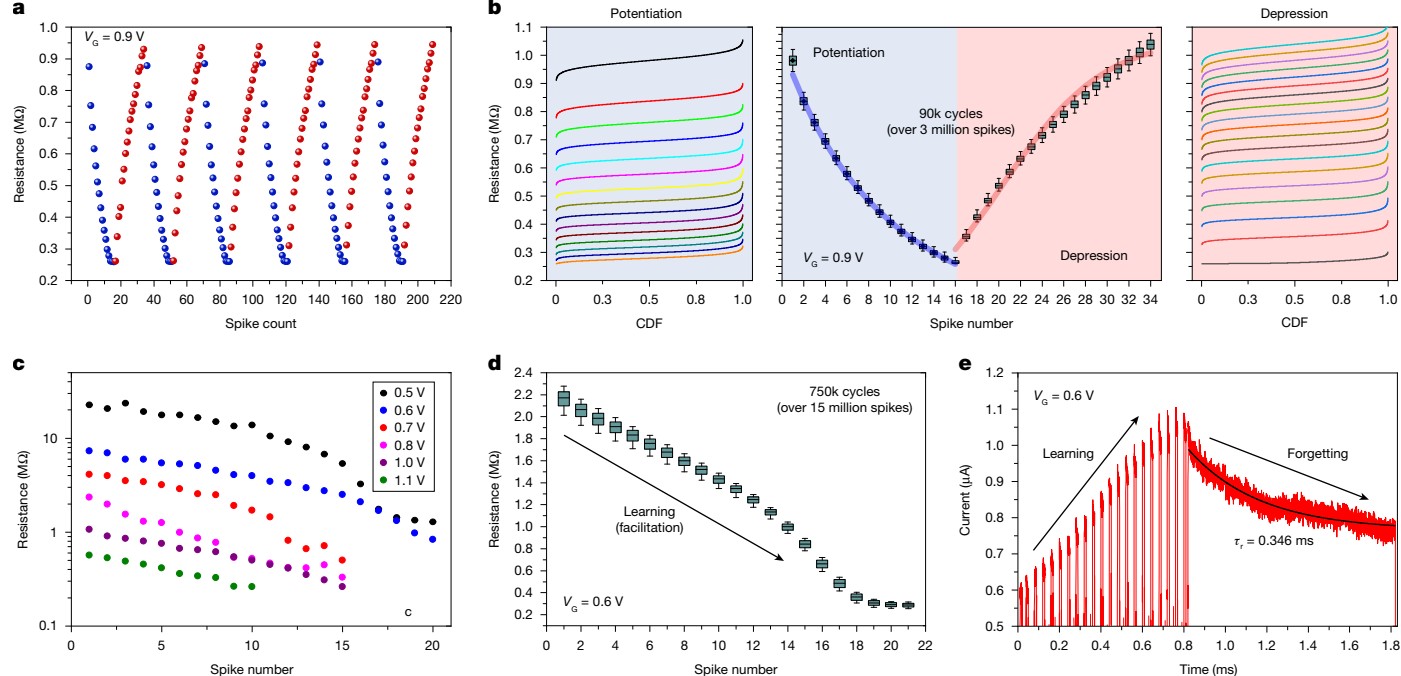

**Fig. 4 | Synaptic plasticity of 500-nm floating-bulk MOSFETs. a**, Detailed view of the synaptic resistance through six potentiation and depression cycles (15 potentiation pulses and 19 depression pulses each). **b**, Statistical analysis of the synaptic resistance under a continuous pulsed regime ($V_{pot} = 4.0$ V, $V_{dep} = -0.4$ V, $V_{read} = 0.3$ V, $t_{pot} = t_{read} = 5$ μs and $t_{dep} = 1$ μs; Supplementary Fig. 23) for 90,000 potentiation and depression cycles (over 3 million spikes). Shown are the empirical cumulative density functions after each pulse of the potentiation (left) and depression (right) cycles. The typical cumulative density function axes are rotated to align with the centre plot. The box plot (centre) shows statistical linearity and state overlap (thick curves are best fits of a simple exponential nonlinear model). Boxes are placed on the median value and represent the 25th and 75th percentiles. Whiskers indicate the 5th and 95th percentiles. Data points per box, 83,734. **c**, Dependence of the synaptic potentiation range under different $V_G$, showing tunability of up to ×20 ($V_{pot} = 5$ V, $V_{read} = 0.5$ V, $t_{pot} = 10$ μs and $t_{read} = 50$ μs). **d**, Potentiation statistics for a succession of spikes (facilitation or learning process) for over 15 million pulses ($V_{pot} = 2.8$ and $V_{read} = 0.75$ V). In each cycle, after the 21st facilitation spike, a negative spike ($V_{reset} = -0.7$ V) reset the synapse to its initial state ($t_{pot} = 5$ μs, $t_{read} = 10$ μs and $t_{reset} = 1$ μs; Supplementary Fig. 24). Boxes are placed on the median value and represent the 25th and 75th percentiles. Whiskers indicate the 10th and 90th percentiles. Data points per box, 771,000. **e**, Detailed view of the synaptic relaxation (forgetting process) when no input spikes were applied after progressive potentiation (learning) in the pulse regime ($V_{read} = 0.75$ V, $V_{pot} = 2.4$ V and $t_{read} = t_{pot} = 10$ μs). CDF, cumulative density function.

times of the order of 1–100 ms (Fig. 4e), well in agreement with the observed dynamics of firing a single pulse (discussed in Fig. 3d) and within the range of synaptic plasticity displayed by biological systems[31] and other emerging devices being explored today[6]. This regime relies on characteristic carrier lifetimes and device capacitances that translate to a precise control of the conductance state and operating range, and it could have an immediate impact on neuromorphic computation strategies, such as recurrent neural networks[33] or reservoir computing[34], where emerging devices (such as memristors) struggle because of their inherent variability.

## Long-term plasticity in a single transistor

Next, we extended our analysis of the floating-bulk transistor when mimicking long-term synaptic plasticity. This feature is used in the widely adopted compute-in-memory accelerators based on hardware-implemented ANNs[10,35,36]. First, we applied $I_D$–$V_D$ sweeps between 0 and 5 V at constant $V_G$ (Fig. 5a). We observed clear hysteresis with different widths at different $V_G = 0.7$, 0.8 or 1 V (Fig. 5a, inset) with a resistance ratio as large as ×12. This ratio is comparable to that of the state-of-the-art metal–oxide and phase-change memristors used for neuromorphic computing[11,37]. Sweeping the voltage to negative values between 0 and −3 V brought the device back to its original state. We analysed the impact of the depression sweep voltage ($V_{dep}$) on the resistance state tuning (Fig. 5b). We observed an approximately ×35 resistance ratio between the high-resistance state ($R_{HRS}$) and the low-resistance state ($R_{LRS}$) (from 15 MΩ down to 400 kΩ, read voltage of 0.5 V; inset of Fig. 5b).

We also assessed the retention at constant read voltage of at least six resistance states after different reset voltages (Methods), achieving long-term synaptic stability over $10^4$ s (approximately 2.8 h; Fig. 5c) without any kind of refresh. This behaviour still occurred when the read-out consisted of 100-ms-long pulses spaced over 900 ms (Methods and Supplementary Fig. 25a,b)—discarding the possibility of the state being maintained artificially by a constant read voltage—and even when tested at high temperatures (85 °C; Supplementary Fig. 25c,d). Also note that we drove this process with pulses, initially in the range 10–100 ms (Supplementary Fig. 26). Therefore, to address the repeatability of this process under a pulse regime that better suits real application conditions, we evaluated the switching performance using 500-μs pulses of +5 V voltages for potentiation, −3 V for depression and 0.35 V for reading the acquired state after each pulse (Fig. 5d). We observed that the devices consistently switched between two distinctive resistance states (ratio more than ×10) for more than $10^5$ cycles (Fig. 5e,f).

This bipolar drain bias regime cannot be employed in grounded-bulk devices, which would have high drain–bulk forward bias currents during the depression sweeps ($V_D < 0$) and no impact-ionization firing during potentiation ($V_D > 0$). In this regime, carrier lifetimes and internal device capacitances cannot account for the observed results. Therefore, to explain the long-term retention, we considered a contribution from carrier injection into the gate dielectric driven by impact ionization in the floating-bulk device. These charge-trapping processes were not a reliability concern for these devices: (1) They were operated within nominal voltages in all conditions (all voltages within 5.5 V). (2) The amount of charge trapped and de-trapped in every cycle did not trigger

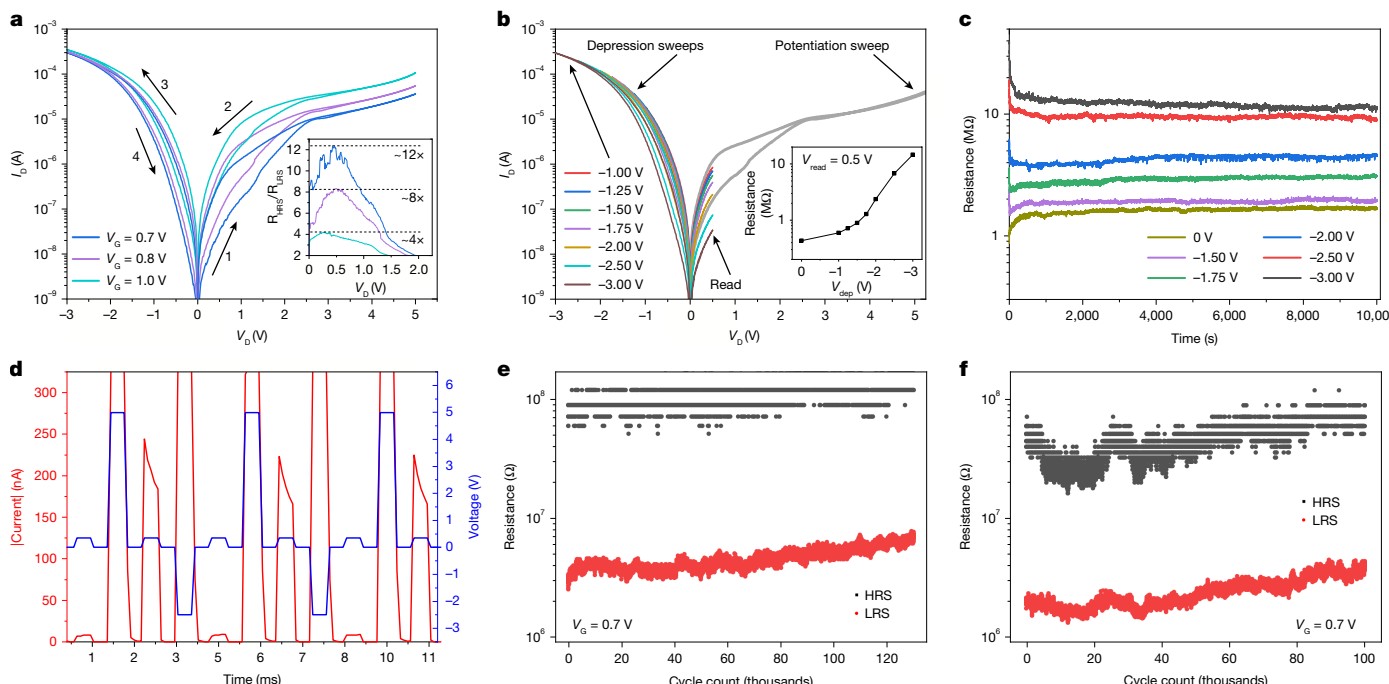

**Fig. 5 | Long-term synaptic behaviour of 500-nm floating-bulk MOSFETs.**
**a**, Quasi-stationary $I_D$–$V_D$ of a transistor in the fully floating-bulk condition displaying bipolar synaptic update. The potentiation sweeps (curves 1 and 2) show different conductance changes at different $V_G$. The depression sweeps towards negative voltages (curves 3 and 4) resets the initial conductance state. **b**, Synaptic update after a potentiation sweep followed by successive depression sweeps stopped at different negative $V_D$ voltages. The conductance state was read after each depression sweep through an $I_D$–$V_D$ curve up to 0.5 V. Inset, read resistance extracted at $V_D$ = 0.5 V for different depression sweep voltages $V_{dep}$.

**c**, Retention of the synaptic update after different $V_{dep}$ sweeps. Each retention measurement was preceded by a potentiation sweep and a depression sweep like those in **a** and **b**. **d**, Current and voltage waveforms showing the synaptic behaviour in the pulsed regime. All pulses were 500 μs wide and $V_{pot}$ = 5 V, $V_{dep}$ = −2.5 V and $V_{read}$ = 0.5 V. **e**, Example of the repeatable synaptic behaviour for more than 100,000 cycles in one device under the pulsed regime depicted in **d**. **f**, Repeated experiment of panel **e** on a separate device, displaying reproducible endurance characteristics.

device failure, even after 100,000 cycles (Fig. 5e,f). This endurance is higher than that of some transistor-based commercial memory technologies, such as charge-trapping transistors (1,000 cycles)[38], and on par with embedded Flash (10,000–100,000 cycles)[39] and some commercial memristor technologies (100,000 cycles)[40]. The results obtained (Fig. 5e,f) for the floating-bulk device agree well with this requirement. We performed TCAD simulations of the floating-bulk MOSFET ($L_{CH}$ = 500 nm) and observed that the conditions were suitable for hot-electron injection during depression and hot-hole injection during potentiation, mechanisms that are used to write and erase commercial floating-gate memory[41] (Methods, Supplementary Note 4 and Supplementary Fig. 27). We validated this experimentally with grounded-bulk measurements of retention and post-retention $I_D$ versus $V_G$ curves (Methods), where we observed sustained retention and a clear subthreshold characteristic shift for each condition (Supplementary Fig. 28).

## Discussion

Standard n-type MOSFETs operating under a floating-bulk condition can mimic several neuro-synaptic behaviours with high performance, good energy efficiency and low variability with high tunability and low area overhead. As all the processes driving these behaviours are controlled by the charge distribution in the semiconductor (with or without charge-trapping), a device operates with outstanding robustness in all regimes. It could be argued that the operating voltages (between 3.6 and 4.5 V in the microsecond-pulsed regime) are a downside compared to standard-operated CMOS transistors, although they are very competitive compared to memristors and other emerging devices (Supplementary Table 1). However, the punch-through

avalanche is a parasitic effect under nominal operating conditions. It can be minimized with specific device engineering (for example, halo or punch-through stop implants[42,43] or layers[44]). Therefore, several device-level parameters can be engineered to enhance specific performance metrics (including operating voltages). As evidence for this, note that all the neuro-synaptic behaviours demonstrated in MOSFETs with $L_{CH}$ = 180 nm have been reproduced in MOSFETs with $L_{CH}$ = 500 nm while applying voltages within their nominal (5.5 V), meaning that they are driven by the same fundamental phenomenon. Therefore, this operating regime should not be a risk to reliability (for a detailed discussion, see Supplementary Note 5).

An attractive characteristic highlighted in our results is the potential for a very simple and compact NS-RAM cell to mimic several neuro-synaptic behaviours. From the neural perspective[8,28], in addition to the threshold voltage modulation, controlling the bulk biasing network can be used for refractory period modulation, for sensitivity tuning of the excitatory or inhibitory signal and for adapting the spike frequency. The same building block can operate in current or voltage mode (Supplementary Figs. 29–32), provide fast leaky-integrate-and-fire or spike frequency adaptation over a wide range of bursting frequencies and input stimuli (Supplementary Note 6), accommodate external reset signals (through bulk bias control pulsing) or operate as a two-, three- or four-terminal neuron in different neural network architectures[45] (Supplementary Note 6 and Supplementary Fig. 33). From the synaptic perspective, a single device could, in principle, replace static random access memory (a volatile memory cell comprising at least six transistors; Fig. 1b) in binarized weight neural networks, or embedded Flash in multilevel synaptic arrays, with the immediate advantage of a significant area and cost reduction per bit. Moreover, the synaptic behaviours presented here could potentially be enhanced in MOSFETs

using metal−oxide high-*k* dielectrics ($HfO_x$), as these materials allow charge-trapping and charge-de-trapping to a much higher degree than $SiO_2$ (as used in this study)[46]. Additionally, the long state retention that we observed in floating-bulk transistors could be sufficient for implementing compute-in-memory approaches with a weight refresh, such as approaches using dynamic random access memory[47,48], but with the advantage of clear long-term synaptic retention through read processes (Fig. 5c), thus providing a substantial improvement in latency. In dynamic random access memory, the term latency is related to the time from when access is requested to when the access becomes possible.

Another important advantage (compared to emerging devices) is that, through well-known device modelling techniques (either physical or compact models), our MOSFET-based neurons and synapses can be immediately introduced into standard CMOS design processes. In contrast to the vastly explored design and implementation methods for ANNs with long-term synaptic devices[10,11,37], the system design process for neural networks based on short-term neuro-synaptic characteristics still lacks mature, well-correlated computation models and network algorithms[6]. In this direction, we have been able to perfectly model the control of the avalanche regime using an open-source SPICE simulator and basic MOSFET and bipolar transistor models. Our results show excellent agreement with the experimental static and dynamic characteristics (Supplementary Fig. 34). This approach may rapidly enable complex system designs and simulations.

In the history of microelectronics, neural and synaptic behaviours based on different physical phenomena have been observed in different types of devices and circuits (Supplementary Note 7 and Supplementary Table 2). In all cases, the time between the first demonstration of a neuron or synapse and the first ANN demonstration (hardware implementation, not simulation) was at least 7 years. For some device technologies (such as ferroelectric FETs), this has still not been achieved. The reason is that moving from a single device to an ANN still requires immense engineering work, including the development of a very large amount of custom-designed peripheral circuits and interfaces with external circuitry for testing and several iterations of circuit-, block- and system-level design, fabrication and test before a functional prototype can be achieved, in addition to the technology integration challenges that different devices may present. However, the adoption of unconventionally biased MOSFETs for mimicking neuro-synaptic responses could be a fundamental breakthrough towards accelerating the next generation of neuromorphic computers without incurring major technological changes. MOSFETs keep surprising us and now−after this study−they seem to be the perfect building block for implementing ANNs.

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

## Methods

### Devices under test

The devices were standard polysilicon/SiON gate-stack bulk-silicon MOSFETs with an n-channel length of 180 nm (thin oxide, approximately 3.5 nm) or 500 nm (thick oxide, approximately 10 nm) from a standard, commercial CMOS technology. All devices had an n–p junction connected to the gate to avoid damage due to the antenna effect during device fabrication, a typical design requirement when large probing pads are directly connected to a thin oxide.

### Basic device characterization

For the current versus voltage ($I$–$V$) curves, we characterized the current flowing through the drain and the source terminals ($I_D$) when the voltage between them ($V_D$) was ramped and the gate electrode was subjected to a constant voltage ($V_G$) while keeping the bulk terminal grounded (as in most applications) or floating. The electrical characterization was performed with a probe station (EPS150, FormFactor) connected to a semiconductor parameter analyser (Keysight B1500A). All the $I$–$V$ curves under d.c. voltages (Fig. 2 and Supplementary Figs. 2–7) were collected using ramped voltage sweeps at limited auto ranging. For constant sweep rate measurements at rates below 100 V s$^{-1}$ (Supplementary Figs. 8, 19 and 14), limited auto ranging was configured starting at a range of 100 nA to ensure a constant delay time between measurements (verified by storing timestamps for every measurement), and the step size was increased (for high sweep rates) or the delay time was extended (for very low sweep rates) to increase the sweep rate parametrically. In all cases, three source measurement units (SMUs) were used for the drain, source and gate of the floating-bulk device and another SMU was used to bias the gate of the substrate current control transistor ($V_{G2}$). The source for this transistor was grounded through the ground unit of the semiconductor parameter analyser. The drain was connected to the bulk tap connection of the floating-bulk device. Retention and pulsed-read retention measurements (Supplementary Figs. 25, 26 and 28) were carried out in Keysight's EasyEXPERT environment using a custom arrangement of $I$–$V$, $I$–$V$ pulse and $I$–$V$ list sweeps. In the floating-bulk and non-floating-bulk retention tests (Supplementary Fig. 26), which addressed the permanent nature of the device state, the substrate tap contact was opened using the built-in SMU switch for the floating-bulk writing procedures and then grounded through the SMU during the retention period.

### Physical simulations

Physical simulations of the floating-bulk transistors were carried out with commercial TCAD software (Sentaurus TCAD, Synopsys). The two-dimensional device structure was built using a structure descriptive approach and based on commonly known parameters of the technology node at which the devices under test were fabricated. Next, the device structure was optimized by simulating the nominal (grounded bulk) quasi-stationary $I_D$–$V_G$ characteristics and calibrated against experimental data. Once good agreement was reached, transient $I_D$–$V_G$ simulations including impact-ionization physics were run to calibrate the impact-ionization model parameters to the nominal substrate currents in the device. With this calibration fixed, we ran transient simulations of $I_D$–$V_D$ at different $V_D$ sweep rates within a mixed-mode environment that connected the bulk biasing network to the substrate contact of the two-dimensional device structure. The bulk biasing network had a constant value resistance ($R_{sub}$) or an n-channel transistor modelled in SPICE BSIM and biased by an independent gate voltage source. In all cases, a bulk connected capacitance ($C_{bulk}$) was included to account for experimental and device connection parasitics. The complete Sentaurus Workbench project, including the command and parameter files, is available through a public repository at https://doi.org/10.5281/zenodo.13843362 (ref. 49). For more details of the models used and the project structure, see Supplementary Note 8.

### Punch-through impact ionization

In the devices under test (Fig. 2), when $V_G > 0.5$ V, avalanching was observed as a slight current increase, like that observed in partially depleted silicon-on-insulator MOSFETs[50]. This effect is often referred to as the kink effect[27]. By contrast, when $V_G$ was between 0.3 and 0.5 V, this phenomenon was manifest as a gate-voltage-dependent hysteresis[51] as wide as 0.5 V, which could be precisely controlled through $V_G$ (Supplementary Fig. 2). Measurements of the grounded-bulk terminal current show that the impact-ionization hole currents were as large as about 10 μA under nominal voltages for 180- and 500-nm devices (Supplementary Figs. 4 and 10, respectively). In the floating-bulk configuration, although electrons were collected by the drain, the excess holes tended to forward bias the source junction (Supplementary Note 3) and introduce a positive feedback during the impact-ionization process, driving the device into an avalanche regime in which the current ($I_D > 1$ μA) was limited only by the spreading resistance of the bulk, with a slight dependence on $V_G$ (through modulation of the depletion region). This universal phenomenon is ascribed to the structure of the MOSFET device, as the same behaviour was observed in floating-bulk thick-oxide MOSFETs (500 nm channel length and approximately 10 nm oxide thickness). In these devices, the hysteresis reached widths of 0.8 V at $V_G = 0$ V and was highly reproducible over several devices with a very low variability (Supplementary Fig. 3) and a wide dynamic range of over 4 orders of magnitude in current, comparable to other threshold devices proposed in the literature as neurons[29,52,53]. Like the results in Fig. 2g for 180-nm devices, this was also a highly repeatable regime for 500-nm transistors, as shown for the more than 70,000 cycles under fast ramped tests at rates of 5,500 V s$^{-1}$, which measured the device current with high temporal resolution (1 μs) at a peak $V_D = 4.25$ V. The result shows a resistance window over ×10 (limited only by the dynamic range of the measurement unit at a fixed amplifier gain) with very low cycle-to-cycle variability (Supplementary Fig. 8).

### Bulk bias control transistor (two-transistor cell)

The effective resistance of a bulk connection to any CMOS device is always determined by device design and processing (such as doping profiles, die size, back-wafer contact surface and top-wafer bulk contact design). Therefore, die-to-die variations can occur during characterization after dicing. Identical 180-nm transistors from different dice can show different hysteretic characteristics depending on variations of the effective resistance of the back-wafer connection. For 30 dice extracted from a multi-project wafer, this was found to depend on wafer position (Supplementary Fig. 5). A bulk control transistor can effectively mask these variations and fix the operating conditions of the floating-bulk device (see detailed characterization of the effective $R_B$ resistance as a function of $V_{G2}$ in Supplementary Fig. 6). This ensures the desired neural behaviour, despite die-to-die (or even wafer-to-wafer) variability, as can be seen from the $I_D$–$V_D$ curves carried out on the same 30 dice but with the control device biased at $V_{G2} = 1.3$ V in all cases (Supplementary Fig. 7).

### Time-domain measurements

All the time-resolved $I$–$V$ curves under pulse or fast ramp modes (sweep rates over 10 V s$^{-1}$) were collected using a two-channel waveform generator and fast measurement unit (Keysight B1530 WGFMU) connected to the drain and source. Two SMUs established a constant bias gate voltage to the device under test ($V_G$) and to the substrate current controlling device ($V_{G2}$) throughout the whole process. These measurements (Figs. 3, 4 and 5d and Supplementary Figs. 15–24) were carried out in a custom environment programmed in MATLAB and C++ running on a personal computer with a GPIB/USB connection to the B1500 mainframe. The environment allowed to rapidly optimize the pulse and ramp parameters (amplitude and timing). We could perform several iterations of long acquisitions, which maximized the use of the instrument

memory, for endurance and cycling tests lasting several hours without user interaction.

## Neuro-synaptic tuning experiments

To extract the firing time and energy, we applied $V_{G2} = 1.6$ V and, by controlling the transistor gate voltage ($V_G$) between 0.35 and 0.45 V (Supplementary Fig. 19a–c), we tuned the neuron firing times between 10 μs and 2 ms for $V_{spike}$ between 3.5 V and 4.5 V (Fig. 3c, left axis). Changing the bulk biasing conditions through $V_{G2}$ had a trivial effect on the firing time within this regime, as shown by the negligible difference between measurements performed at $V_{G2} = 1$ or 1.6 V (Supplementary Fig. 19d). This behaviour was also controlled using oxide transistors that were 500 nm in length and thickness (Supplementary Fig. 20), for which equivalent dynamics was observed but at lower currents (down to 10 nA) and slightly higher voltages (up to 5 V). This is of interest for the device-level optimization of the neural behaviour through design variables, such as transistor size, oxide or threshold voltage, which is typically offered in standard CMOS processes.

To characterize the leaky feature (Fig. 3b and Supplementary Fig. 21), we applied a single voltage pulse of 4.5 V for 1 ms to fire the neuron. The relaxation transient was extracted at a constant read voltage of 1 V. This time, by using different $V_{G2}$ values (from 0 to 1.8 V) at constant $V_G$, we showed that the relaxation transient can be tuned from approximately 50 μs to tens of milliseconds. A long-term change of the resistance from a HRS to a LRS was visible (Supplementary Fig. 21d). The characteristic relaxation time ($\tau_r$) was extracted through an exponential decay fit, and the synaptic update ratio was evaluated after a 30-ms window ($R_{HRS}/L_{LRS}$, as depicted in Supplementary Fig. 21d).

To mimic biological processes, neuron devices need to show a characteristic responsiveness to frequencies that are typically in the range 20 Hz to 20 kHz (for example, for frequency mapping of audio signals). To assess this, we used a train of spikes of fixed duration ($t_{spike} = 5$ μs) and different amplitudes ($V_{spike}$ between 3.6 and 4.5 V) and frequencies (between 20 Hz and 200 kHz) at a fixed $V_G = 0.4$ V and $V_{G2} = 0.8$ or 1.3 V, thus covering different relaxation dynamics. Under these conditions, we extracted the time elapsed until the neuron fired at each given input (Supplementary Fig. 22) and parametrically mapped it in Fig. 3d. The response can be tailored according to the system needs and to the process being mimicked. For tonotopic mapping[29], different devices can be biased to provide a specific firing time at different frequencies to provide a full range of audio signal responses that spans from the lowest audible frequencies to signals well into the ultrasound range, such as for the efficient implementation of a smart hearing system. The wide configurability of the neuron can find application in various general-purpose neuromorphic implementations.

## Charge-trapping mechanisms in long-term synaptic behaviour

For a detailed discussion, refer to Supplementary Note 4 and figures therein. From the theoretical aspect, it is probable that the injection of hot electrons during the reset (negative drain bias) in the floating-bulk condition contributes to the increase of the threshold voltage. Moreover, de-trapping some of this injected charge or hot-hole injection may be the mechanism through which the threshold voltage is reduced back to its initial value. In the reset process (the increase of the threshold voltage or, in other words, the increase of the resistance under a constant bias), a negative bias is applied to the drain in the floating-bulk condition. Note that if the transistor bulk was indeed grounded, the current would be determined by the forward bias drain–bulk junction and would rise rapidly, as it would be limited only by the semiconductor spreading resistance and the interconnect resistance (see measurements and TCAD simulation results for these conditions in Supplementary Fig. 27a). However, with the floating bulk, the decreasing drain voltage tends to forward bias the drain–bulk junction, lowering the electrostatic potential of the silicon bulk and inducing an inversion channel under the gate (recall that $V_G$ was held at constant voltage).

As the source was held at 0 V, it was biased above the electrostatic potential of the bulk and large currents were driven in the device channel. When the drain voltage was sufficiently negative, there were energetic electrons in the vicinity of the source terminal (see impact-ionization rates from the TCAD calculations in Supplementary Fig. 27b), and these were probably injected into the gate oxide. This would, logically, result in an increase of $V_{th}$ and therefore a reduction in the current drive capability. This would tend to shift $V_{th}$ to a higher voltage, therefore lowering the drive current capability of the transistor, which translates to a high resistivity state under read conditions.

During the set process, some de-trapping of the charge generated during the reset sweep would be expected, but the effect of hot electrons is typically non-reversible[54] (at least without annealing conditions). Therefore, it is probable that the injection of holes through the gate dielectric on the drain side may also take place under impact-ionization conditions. This well-known process has been observed in standard silicon transistors since 1981[55–57]. This phenomenon takes place at high drain voltages, where band-to-band tunnelling is likely in the drain–bulk junction and highly energetic holes and electrons are present under impact-ionization conditions. As excess holes are not collected by the bulk current in the floating-bulk condition and the density of holes tends to increase at the oxide interface close to the drain (as discussed previously in Supplementary Fig. 12 and Supplementary Note 3), the conditions are suitable for hole injection. As with electrons, hot holes can be injected through the gate dielectric if they have enough energy to overcome the energy barrier. As a result, this effect has been observed to be responsible for the read disturb instability in EEPROM[58,59] and is employed as an erase mechanism in some commercial embedded Flash memories[39], as injected trapped holes result in a decrease of the threshold voltage. In such cases, the hot holes that are injected can be effectively concealed within the floating-gate structure, but in standard MOSFET structures, fewer holes can become trapped in defect centres of the gate oxide or spacer oxide of the MOSFET structure.

## Short-term plasticity experiments

In Fig. 4a, we applied pulse trains with a constant amplitude in groups of 15 potentiation pulses ($V_{pot} = 4.1$ V and $t_{pot} = 5$ μs) followed by 20 depression pulses ($V_{dep} = -0.25$ V and $t_{dep} = 1$ μs). This set of potentiation and depression pulses was intercalated by a read pulse ($V_{read} = 0.3$ V and $t_{read} = 5$ μs) that measured the resistance of the device. For Fig. 4b, we tuned the amplitudes ($V_{pot} = 4.0$ V, $V_{dep} = -0.4$ V and $V_{read} = 0.3$ V) and applied this protocol continuously for approximately 200,000 cycles (approximately 7 million pulses; Supplementary Fig. 23). We extracted statistics for the obtained synaptic weight after each pulse (Supplementary Fig. 23a). In Fig. 4d, the potentiation process consisted of 21 potentiation pulses at a lower potentiation voltage ($V_{pot} = 2.8$ V and $V_{read} = 0.75$ V) followed by a single depression pulse ($V_{dep} = -0.7$ V; Supplementary Fig. 24). This process allowed us to rapidly reset the neuro-synaptic characteristic of the device to a quiescent initial state. We observed that the depression pulse effectively reset the synapse to its initial weight, which was tunable over a window of approximately ×10, following a roughly bilinear characteristic (Fig. 4d). The observed cycle-to-cycle variability was ascribed to drift in the time-domain measurement and some degree of probe-to-pad contact stability in pulsed experiments spanning several hours (760,000 cycles or 16.7 million pulses; Supplementary Fig. 24). In Fig. 4e, the potentiation sequence (learning) was the same as for Fig. 4d, whereas the synaptic decay through time (forgetting) was performed under a slightly lower constant read voltage ($V_{read} = 0.7$ V).

## Neuron bursting, spike frequency adaptation measurements

Bursting-mode neuron measurements (Supplementary Figs. 29–31) were carried in the same probe station set-up by forcing a current through the SMU connected to the drain of the floating-bulk device. This terminal was split and fed to the input of a high-input-impedance

low-noise voltage follower (TLC2262). Its output drove a channel of a digital sampling oscilloscope (MSO-X 3024G, Keysight). The oscilloscope–semiconductor parameter analyser tandem was synchronized using a MATLAB script and the whole parametric space of $V_{G1}$, $V_{G2}$ and $I_{excitatory}$ was swept. The action potential bursts were captured in each condition (Supplementary Fig. 31a). Further details and results are given in Supplementary Note 6.

## Data availability

The minimum dataset necessary for interpreting, verifying and extending this research is available at Zenodo (https://doi.org/10.5281/zenodo.13843362)[49].

## Code availability

All code related to TCAD simulation projects, SPICE models and SPICE simulation net lists is publicly available at Zenodo (https://doi.org/10.5281/zenodo.13843362)[49].

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

**Acknowledgements** This work has been supported by the Baseline funding scheme of the King Abdullah University of Science and Technology, and by the Startup Fund provided by the National University of Singapore. Prof. Mario Lanza acknowledges the platform Web of Talents (https://weboftalents.com) for support in the recruitment of students and postdocs.

**Author contributions** S.P. and M.L. designed the experiments. S.P. designed the CMOS devices, characterized the microchips in all conditions and performed the TCAD simulations. M.A.V. provided simulation insights for the SPICE model and TCAD simulations. K.Z., O.A., W.Z., Y.S, Y.Y. and Y.P. fabricated the customized electrodes used to characterize the microchips. S.P. and M.L. wrote the manuscript, which was revised by all the authors.

**Competing interests** The authors declare no competing interests.

**Additional information**
**Correspondence and requests for materials** should be addressed to Mario Lanza.
