## [Peer Review File · Nature]

Synaptic and neural behaviours in a standard silicon transistor

Corresponding Author: Professor Mario Lanza

This file contains all reviewer reports in order by version, followed by all author rebuttals in order by version.
5`gc`fYj JYk Yf`%j Yfgjcb`%Vt`a a Yblg`D8 : `jg`UXXYX`UhH`Y`YbX`cZl YfYj JYk`
Parts of this Peer Review File have been redacted as indicated to remove third-party material.

Version 0:

Reviewer comments:

Referee #1

(Remarks to the Author)

A. Summary of key results. The authors propose and demonstrate experimentally that a single MOSFET can be controlled to operate in the whole range of bio-mimicking devices, from neurons to synapses. Using a well-know device such as the MOSFET (a single device) for ll these purposes would certainly be a breakthrough

B. The proposal of this paper is fully novel to the best of my knowledge.

C. I don't see any problem with the experimental methodology. However, I suggest the authors to perform physics-based simulations to confirm their interpretations of the device behavior under different regimes of operation. (see concrete proposal in the attached pdf file)

D. The required statistics are sufficiently reported.

E. Conclusions. Provided that the theoretical interpretations are correct, the conclusions (based mainly on experimental results) are valid. Please see detailed report in the attached pdf file.

F. Suggested improvements. Please see details in the attached report in pdf file. The description of experimental results is in my view adequate. However, I suggest to make some additional measurements of the retention of stable states, unless the authors justify that this is not really necessary.

G. References and fully appropriate to the best of my knowledge

H. The manuscript is very clear and it is nicely contextualized with respect to the state-of-the-art. Abstract, introduction and conclusions are also ok. I make some comments in the attached file to eventually modulate the conclusions.

Referee #2

(Remarks to the Author)

This paper investigates using standard silicon CMOS transistors to replicate electronic neural and synaptic behaviors when what the authors call biased unconventionally. The authors propose this method as the feasibility of a potential short-term solution for implementing neuromorphic circuits in artificial intelligence systems while considering the challenges associated with traditional MOSFET and CMOS-based approaches. While the authors have provided lots of experimental results from two device types (nMOS with two different oxide thicknesses), the results are not fully surprising as the kink effect from Figure 1b is also observed when the IDVG of FDSOI devices is investigated [1]. The kink the authors see in Figure 1b originates from the impact ionization effect, which the authors explain. However, the bulk must remain floating to see such a kink, similar to FDSOI. By adding a transistor to the bulk contact to mimic a series resistor with variable resistance – from high R when VG2 is 0V to low R when VGS2 is above the Vth of the bulk transistor – the authors can switch between an FDSOI device and a conventional nMOS transistor, where the bulk is typically grounded.

Regarding the device, the authors mention that the channel length is 180nm and 500nm. Still, it would also be interesting to know which oxide thickness has been used for the investigated technology. As VD increases to 3V and higher, I assume an

EOT exceeding 7nm. The oxide thickness is an important parameter here as the show effect relies on impact ionization, and here, the oxide has to sustain the high VGD. Consequently, the synopsis behavior will only be achievable for a technology with a minimum EOT, which I assume exceeds the values of highly scaled nodes. This might be a showstopper for the investigated effect.

Another issue I see in this work (an issue for high novelty requested by Nature Journal) is that the synopsis behavior is instead a binary switching between LRS and HRS rather than an analog synopsis. This leads to limitations in the precision of synaptic weight representations, and neural networks from binary synopsis might need help to capture complex patterns. Also, other simple and established circuits are available to mimic binary synopsis, e.g., SRAM cells, which can be scaled very well and operate the transistors within VDD, thus ensuring a more extended device and circuit lifetime. The operation within the nominal VDD also enables the SRAM cell to be scaled; however, as the approach shown in this work requires unconventional high biases, the scaling options might be limited. Thus, these devices might need more chip area.

Although the work has been carefully carried out, the main focus was on describing a phenomenon of a single silicon transistor. As Si devices are pretty mainstream, I expect the authors to use the proposed effect to demonstrate a neural network on a chip.

As a last point, the authors assign the synaptic behavior to a single Si transistor with unconventional biasing in the abstract. However, strictly seen, this needs to be corrected as at least two transistors are required (an additional one to tune RB).

[1] https://theses.hal.science/tel-02506292v1/file/PARK_2019_archivage.pdf (page 34, Figure 2.1)

Referee #3

(Remarks to the Author)

With this paper the authors tackle the problem of area over-head of state-of-the-art neuron and synapse circuits by proposing a 2-transistor cell circuit that if biased properly can show: a. neuron and synapse dynamics and b. synaptic plasticity mechanisms. The authors propose to operate the device on the verge of punch-through conditions exploiting the resistance of the bulk connection to ground to switch between neuron, synaptic and plasticity mode.

The authors show that this 2-transistor cell can reliably produce multiple neuro-synaptic behaviors.

To the best of my knowledge the papers shows an original approach to realize neuromorphic circuits and it could significantly drive to the field to more efficient and smaller implementations. However, it is not clear from the Introduction whether this is the first attempt to use the transistor in this regime for neuromorphic applications or not. If it is, the authors should state it more clearly. If it is not, the authors should cite related references.

The paper is well written, and the results are clearly explained. Multiple neuron-synaptic behavior has been reported such as threshold voltage neuron modulation, firing capabilities from 20Hz to 200kHz, synaptic potentiation and depression and retention of the synaptic update up to 2 hours. Moreover, a control on the bulk bias of the MOSFET of the 2-transistor cell circuit enables robust and reliable results.

I have no suggestions for improvement on the current content of the paper, but I would like the authors to address my following comment/curiosity.

It would be interesting if the authors could extend the last part of the discussion and explain how they would integrate these 2-transistor cell in a large-scale network where neurons are connected to groups of synapses. If the idea is to use the same 2-transistor cell to implement synapses and neurons, how would the synaptic current excite the neuron circuit. How would the current from the neuron circuit be read out and how would the communication between different neurons be handled?

Version 1:

Reviewer comments:

Referee #1

(Remarks to the Author)

A. Summary of key results

The authors demonstrate that a single conventional MOSFETs can be used to implement neurons and (tunable) synapsis. This would certainly be a great advantage for the implementation of neuromorphic circuits in conventional CMOS technology. The experimental results are very clear and complete, covering the full range of bio-mimicking devices. In fact, using a transistor connected to the bulk terminal, the authors show that it is possible to modulate the punch-through avalanche mechanism in the transistor to show either an abrupt thresholding behavior (neurons), a configurable time-dependent synaptic plasticity behavior or a synaptic behavior.

B. Originality and significance

The paper is fully novel to the best of my knowledge. The author's proposal seems to be disruptive for the implementation of neurons and synapses using a single conventional MOS transistor. The fact that the authors deal with a conventional "deterministic" device based on well-known physics is certainly a strong advantage over other less conventional proposals. In this regard, the single weak point is that the long-term synaptic behavior is controlled by charge trapping in the gate insulator. In any case, my recommendation is to publish the paper because it is really novel and significant.

C. Data and methodology: validity of the approach, quality of data, quality of presentation.

The authors present a wide variety of data, covering all fundamental aspects of the proposal. In my opinion, their approach is adequate, the presented data is what is required for the purpose of the paper and the results are clearly presented. As in the previous manuscript, I don't see any issue with the experimental methodology.

I also must point out that, as I suggested in my previous review, the authors have complemented their experimental data with TCAD simulations (and not only with a compact model). I think that this has substantially improved the quality of the paper.

D. Appropriate use of statistics and treatment of uncertainties.

The required statistics are sufficiently reported.

E. Conclusions: robustness, validity, reliability

The experimental results cannot be debated and constitute the main piece of contribution of the paper. The interpretation is also solid though sometimes it is not easy to explain all the details through the analysis of the device basic operation, and elements such as traps in the insulator or parasitic capacitances are invoked. I am not raising a criticism here, I am only saying that the involved physics are complicated. In summary, my opinion is that the author's conclusions are robust, valid and reliable.

F. Suggested improvements: experiments, data for possible revision

In my previous revision I made several suggestions and the authors have been able to fully fulfill my expectations. In particular, the most important ones are:

- I raised the question of long-term retention of several states in the synaptic regime that should include some kind of trapping in the device. The authors have described the process in terms of trapping in the oxide.

- I suggested some measurements with different ramp rates in the neuron regime. The authors have made these measurements and provided an understandable discussion of the results.

- I suggested to complement the experimental results with TCAD simulations and they have included these results in this second version of the manuscript.

As far as I can see, I also consider that the authors have adequately replied to the other reviewers' reports.

I have no further comments or suggestions and I recommend publication.

G. References: appropriate credit to previous work?

To the best of my knowledge, the references are fully appropriate.

H. Clarity and context: lucidity of abstract/summary, appropriateness of the abstract, introduction and conclusions.

The manuscript is very clear and nicely contextualized with respect to the state-of-the-art. Abstract, introduction and conclusions are also OK for me. In fact, after revision of the first manuscript, the conclusions are stronger and more reliable.

Jordi Suñé
Universidad Autónoma de Barcelona (Spain)

Referee #2

(Remarks to the Author)

The reviewer would like to thank the authors for their extensive answers to my concerns and updating their manuscript accordingly. There are no more concerns from my side.

Referee #3

(Remarks to the Author)

I would like to thank the authors for addressing my questions. I have no further comments.

We would like to deeply thank the reviewers for their thorough assessment of our work and for the positive feedback on it. Following their recommendations, we have conducted multiple additional experiments, provided additional data, and revised text and figures addressing their comments. In the following, we provide point-by-point answers to their comments, and indicate how they have been addressed in this revision. All responses are provided in-line in blue fonts, together with specific figures (referred in the format Fig. R1, R2, R3, etcetera) and references (found at the end of this letter under the format ^[RL1, RL2, RL3, etcetera]). For clarity, the sections quoted from the original or revised version of the manuscript files are shown *in italics* and revised content is *highlighted in yellow*.

Referee #1 (Remarks to the Author):

A. Summary of key results. The authors propose and demonstrate experimentally that a single MOSFET can be controlled to operate in the whole range of bio-mimicking devices, from neurons to synapses. Using a well-know device such as the MOSFET (a single device) for all these purposes would certainly be a breakthrough.

B. The proposal of this paper is fully novel to the best of my knowledge.

C. I don't see any problem with the experimental methodology. However, I suggest the authors to perform physics-based simulations to confirm their interpretations of the device behavior under different regimes of operation. (see concrete proposal in the attached pdf file).

D. The required statistics are sufficiently reported.

E. Conclusions. Provided that the theoretical interpretations are correct, the conclusions (based mainly on experimental results) are valid. Please see detailed report in the attached pdf file.

F. Suggested improvements. Please see details in the attached report in pdf file. The description of experimental results is in my view adequate. However, I suggest to make some additional measurements of the retention of stable states, unless the authors justify that this is not really necessary.

G. References and fully appropriate to the best of my knowledge.

H. The manuscript is very clear and it is nicely contextualized with respect to the state-of-the-art. Abstract, introduction and conclusions are also ok. I make some comments in the attached file to eventually modulate the conclusions.

We sincerely thank the referee for the thorough revision of our work and, for the positive evaluation, and for the constructive recommendations. We have performed new experiments and simulations, and provide responses for all the comments one by one; the modifications made to improve the manuscript are also indicated.

This is really a very complete, novel and interesting paper. Being able to use single conventional MOSFETs as neurons and (tunable) synapsis would certainly be a great advantage for the implementation of neuromorphic circuits. The experimental results are very clear and complete, covering the full range of biomimicking devices. In fact, using a transistor connected to the bulk terminal, the authors show that it is possible to modulate the punch-through avalanche mechanism in the transistor to show either an abrupt thresholding behavior (neurons), a configurable time-dependent synaptic plasticity behavior or a synaptic behavior. The fact that the authors deal with a conventional “deterministic” device based on well-known physics is certainly a strong advantage over other less conventional proposals.

In general, I agree with the authors discussion but, there is an important point that I am not able to fully understand and which generates me some significant doubts (at least, as far as the interpretation is concerned). My main doubt is related to the behavior of the devices as synapses. The authors show that their devices can show long-term and short-term plasticity (LTP and STP). In this regard, I can qualitatively understand the STP behavior (although the relaxation times are so long that the underlying physics are also not fully clear to me). On the contrary, in the extreme case of fully floating bulk ($V_{G2}=0$), **I have serious doubts about the physical interpretation of the LTP behavior in these MOSFETs.**

We deeply thank the reviewer for raising this important matter. We have addressed this comment in detail in the answer to reviewer comments #1 to #3 further along, for the sake of the organizational clarity of this response letter.

Main comment/objection

My main objection is related to the role of charges accumulated in the bulk of the transistor, particularly when they behave as synapses.

I can understand that during punch-through/avalanche there is a buildup of charge in the body (and in the channel) that changes the potential distribution in the device. This can certainly explain the volatile behavior observed when the devices behave as neurons. This is observed when the backgate transistor resistance is small and favors carrier collection. In this case, when the drain reaches a threshold voltage (which depends on the ramp rate), punch-through and impact ionization are triggered and the current suddenly increases. When V_{DS} is reduced after this punch-through/snapback transition, the potential distribution has changed and lower values of V_{DS} can sustain impact ionization and hence the accumulation of charges. This accumulation of charges disappears when V_{DS} reaches the reset voltage. This explains why there is a memory window. It is also true that in this case, the bulk connection favors the collection of carriers. In this case, I can understand the role of accumulated charges (mainly holes) because all the effects are observed under high drain bias and there is a continuous carrier supply (from impact ionization in the drain).

Reviewer 1 comment #1

Although I can qualitatively understand the role of charges when the devices are operated as neurons, there is one detail that I would also like to clarify for this operation mode. If the V_{DS} backward sweep of (Suppl. Fig. 4a/b and main text page 3 line 163), the reset voltage is supposedly related to recombination of carriers in the parasitic bipolar transistor base (collection from the bulk as well). This is explained in supplementary note 1 page 4, lines 124-130 and in other places of the manuscript. However, no dependence on the sweep rate is observed during the backward sweep. This seems to be in contradiction with the fact that the transition is related to the relaxation of excess of carriers and the return to “equilibrium”, one would expect that faster sweeps would lead to smaller reset voltages because the carrier needs a certain time to relax. **Can the authors clarify this point?**

We thank the reviewer for this interesting comment. For clarity, we have performed new experiments aimed to analyse the impact of the sweep rate on the dynamics of the hysteresis loops. We have also performed detailed TCAD simulations of the devices used in this study, replicating the device structure based on the available information of the process and some general assumptions given by the technology node, such as doping profiles which are not precisely known (these are technology proprietary information that is not available). The Methods section of the manuscript has been revised with a description of the simulation models and strategies used, alongside a link to a repository containing the full TCAD project for public access and a reference to a new Supplementary Note 8 providing complete details of the simulation procedure.

To obtain representative results we first calibrate the models and simulations to the experimental data, particularly to the impact ionization currents observed during the measurements of the standard-biased n-channel transistors, i.e., with bulk grounded (see Fig. R1 comparing TCAD data with experimental I-V curves).

Fig. R1 | TCAD modelling results of the fundamental 180 nm transistor structure. The simulation models and structure are optimized to replicate the main driving mechanism of the floating bulk device, which is impact ionization (characterized by bulk current I_B). TCAD simulation results in transient regime (replicating measurement conditions) show excellent agreement with measurement results. In measurements and simulations, $V_D = 1V$.

To explain the results in full extent, let us split the process in two parts, as depicted in Fig. R2: I - the process when the neuron fires (forward sweep in the I-V), and II - the process that takes place when the neuron turns off (backward sweep in the I-V).

Fig. R2 | ON and OFF transitions of the punch-through impact ionization firing mechanism. a, Turn ON transition (V_{fire}) during ramp up of V_D . **b,** Turn OFF transition (V_{relax}) during ramp down of V_D . In both cases, the SMU is configured in LIMITED (1 nA) AUTO range mode, which results in an average sweep rate of 0.65 V/s.

I – Firing (turn ON)

As drain voltage is increased (ramped up) the build-up of excess carriers related to the impact ionization^[RL1] process will result in the progressive firing of the parasitic bipolar transistor in the structure of the MOSFET (see Fig. 2a).

The excess electrons get drifted/diffused alongside the channel electrons, while the excess holes can do one out of two things. On one hand, excess holes tend to accumulate in areas with the lowest potential near the channel region. This has been well covered by Moselund et al.^[RL2] while exploring punch-through avalanche devices for capacitorless DRAM memory applications. We carry out new technology computer aided design (TCAD) simulations provided as additional information to this letter and as part of revised Supplementary Note 3 and observe a consistent behaviour. Fig. R3 displays the hole density in the structure of the 180 nm MOSFET at equilibrium and fired conditions before and during the firing event.

Fig. R3 | Hole density in the floating bulk transistor. Hole density near the drain region under **a**, equilibrium and **b**, fired conditions. The white lines represent the depletion region. The fired conditions are $V_G = 0.35$ V, $V_D = 3.5$ V, and effective $R_B = 50$ M Ω . **c**, Hole density across the depth of the structure (X) at $Y = 0.07$ μm , displaying the hole excess pocket near the gate oxide interface, caused by an electrostatic potential well.

On the other hand, holes are also collected through the bulk terminal of the MOSFET, namely I_B . This collection process can change substantially with the bulk effective resistance, and hence it is directly affected by the bias of the bulk control transistor V_{G2} . The raise of I_B forward biases the bulk-source junction of the device, which is coincidentally the base-emitter junction of the parasitic bipolar transistor. The fact that the MOSFET is operating near the punch-through condition, means that the effective base of this

bipolar transistor is very narrow, which translates to a high DC current gain, defined as the ratio between base current and collector current ($\beta_{DC} = I_C/I_B$). Therefore, the increase in I_B eventually turns on the parasitic bipolar transistor, inducing a high current that adds to the MOSFET channel current, but that can be orders of magnitude higher (specially while operating the MOSFET in the weak/moderate inversion regions, as is the case in our implementation). At this point, the transistor connected to the bulk of the floating body device plays a crucial role on determining the neuron firing voltage: a higher effective resistance of the substrate means a larger voltage drop due to I_B , hence a higher forward bias of the emitter-base junction of the parasitic bipolar transistor. In other words, the bipolar transistor is “easier” to turn ON, and therefore the voltage value V_D at which the neuron fires reduces when V_{G2} is lower. This is clearly depicted by new measurements of the firing voltage dependence on V_{G2} , conducted at constant V_G and constant ramp rate (see Fig. R4a and R4b), and further verified through TCAD simulations where the increase of bulk-connected resistance translates to a lower onset potential of the firing process, which coincides with a collapse of the depletion region at the source-bulk (emitter-base) junction due to avalanche-induced forward bias (see Fig. R4c and R4d).

Fig. R4 | ON transition voltage dependence on V_{G2} (R_B). **a**, Turn ON transitions during ramp up of V_D for V_{G2} between 0 V and 1.6 V (dashed curve is for a fully floating bulk condition, no device connected to the bulk terminal). Ramp rate is constant at 0.25 V/s in all cases, and a 10 second reset is applied between sweeps by setting $V_{G2} = 5$ V with all device terminals grounded. **b**, V_{fire} dependence on bulk control V_{G2} . **c**, Simulations replicating the conditions from panel **a** in a mixed-mode environment, using a fixed resistance R_B connected to the bulk terminal of the device structure. Left axis shows I_D and right axis shows V_B , displaying the bipolar transistor firing and showing excellent agreement with measurements. **d**, Electrostatic potential contours right before (top) and after (bottom) the device fires.

Once the parasitic bipolar device has fired, a positive feedback loop takes place which self-sustains the base current: the raise of the substrate potential through R_B , which in turn forward-bias the bulk-source

(base-emitter) junction, further increases the drain current through bipolar amplification effect, sustaining the impact ionization condition. Commonly, this is an unwanted parasitic effect which can be effectively reduced by multiple device level optimization steps from the fabrication side^[RL1,RL3-RL5]. But it is also well known that a reduced effective bulk resistance naturally diminishes this feedback effect, which is accomplished in VLSI design by the redundant placement of substrate contacts and the careful design of the bias network on chip.

For measurements performed at very low V_{G2} values < 0.8 V, I_B is only limited by leakage currents of the experimental setup and the external control device (see Supplementary Fig. 6 in the revised version). This is observed as a saturation in the threshold voltage V_{fire} to ~ 2.1 V. If the bulk is fully floating, the minimum observed V_{fire} is ~ 1.8 V (for $V_G = 0.3$ V, see Fig. R4a,b).

II.a – Relaxation at high V_{G2} / low R_B (abrupt turn OFF)

In all cases, when the voltage is swept back down, the current remains high even after the voltage V_D drops below the threshold that was required to attain sufficient impact ionization as to fire the self-sustained bipolar device. This gives origin to the clearly observed hysteresis. At this point, the high current is only sustained through the previously mentioned bipolar effect: as long as the voltage drop in R_B is high enough to keep the bipolar device in direct active mode (amplification), the channel current will remain high (as the product of βI_B), mainly driven by this parasitic device.

However, as the drain voltage is driven lower, two effects take place: (i) the depletion region of the drain will tend to get smaller due to the reduction of the reverse bias in the drain-bulk (collector-base) junction, increasing the effective width of the base of the bipolar transistor and reducing its gain β_{DC} (this is denoted by the slope of the I_D - V_D characteristic in the post-fired condition in Fig. R2a, R2b and R4a and R4d); (ii) the excess carriers provided by the impact ionization process are also reduced, which also represents a decrease of I_B and therefore a weaker forward bias of the emitter-base junction in the bipolar device as excess carriers are collected and the electrostatic potential in the bulk of the device is decreased. All in all, when I_B falls below the threshold value to keep the bipolar device ON, the parasitic device turns OFF, but this process can have very different dynamics depending on how fast the voltage of the bulk terminal can drop. At high $V_{G2} > 1.4$ V, the impedance connected to the channel is relatively low in terms of the voltage drop that is generated due to impact ionization currents (100 nA ~ 1 μ A I_B would generate voltage drops ~ 100 mV). In these conditions, the turn off dynamic is very abrupt (substrate voltage can drop rapidly through a relatively low impedance) and the turn off voltage (V_{relax}) shows negligible dependence on the sweeping rate (see Fig, R6a-c. Meanwhile, the turn ON voltage (V_{fire}) is slightly more sensitive to the sweep rate (see Fig, R6d-f). These differences should be ascribed to the different time dependence of the processes driving the turn ON and turn OFF processes (impact ionization and excess carrier decay/substrate voltage drop, respectively).

II.b – Relaxation at low V_{G2} / high R_B (slow turn OFF)

When the effective bulk resistance is sufficiently high, the presence of parasitic capacitances at the bulk connection can play a major role on the dynamics of the relaxation process. The total parasitic capacitance at the bulk connection will maintain the high potential of the floating body as it slowly discharges, therefore the forward bias of the emitter-base junction is sustained as V_D is swept down, keeping the bipolar transistor ON and showing a less abrupt turn OFF in the backward sweep. Under these conditions, even though excess carriers will tend to recombine (mostly through Shockley-Read-Hall and Auger dynamics) when impact ionization rate is diminished, the capacitance on the bulk acts as a source for excess carriers, maintaining the bulk potential high even as V_D is ramped down. This is clearly represented by the measurements displayed in Fig. R5: at $V_{G2} = 1.4$ V, a very shallow dependence of V_{relax} with the sweep rate can be seen, becoming much pronounced at $V_{G2} = 1.0$ V. In Fig. R6 the behaviour of V_{fire} and V_{relax} as a function of sweep rate is displayed for a wider V_G, V_{G2} space. V_{relax} shows a clear dependence at low $V_{G2} < 1.4$ V which is much shallower for V_{fire} .

Fig. R5 | Sweep rate dependent hysteresis loops at different V_{G2} . Characteristic I_D - V_D hysteresis sweeps at different sweep rates ranging from 25 V s^{-1} to 0.1 V s^{-1} for two different V_{G2} conditions: **a**, $V_{G2} = 1.4 \text{ V}$, showing very shallow dependence of V_{relax} on sweep rate; **b**, $V_{G2} = 1. \text{ V}$, showing very clear dependence of both V_{fire} and V_{relax} on sweep rate. In both cases, $V_G = 0.35 \text{ V}$.

Fig. R6 | Transition voltages as a function of I-V sweep rate. V_{relax} (top) and V_{fire} (bottom) dependence on I-V sweep rate at different V_{G2} and at V_G of 0.25 V (left column), 0.35 V (centre) and 0.45 V (right column). Note that dependence on sweep rate vanishes (within the measurement range) at higher V_{G2} .

This asymmetry allows controlling the plasticity of the neuro-synaptic device or, in other words, how long it takes for the device to relax back to its stationary relaxed state. It is important to highlight that under very high bulk resistance conditions, the relaxation dynamics will be very impacted by parasitic capacitances, which can lead to significant variability when reproducing this effect amongst different measurement setups or circuit configurations. This can be observed through mixed-mode TCAD transient simulations where a lumped capacitance is connected to the bulk terminal of the device. At relatively large capacitances of 10 pF , it is possible to observe this effect and a good agreement with experimental measurements (see Fig. R7). For high density applications optimizing speed, the slowest that this process can take place will depend on the parasitic substrate capacitance of the device itself and on the effective resistance of the substrate connection (at constant parasitic capacitance, larger resistance translates to slower relaxation of the floating bulk). In the case of applications that prioritize mimicking of biological processes and their time-domain

characteristics (typically slow process with characteristic times > 1 ms), the substrate network can be designed specifically, incurring in area overheads to increase bulk connection resistance and/or capacitance (a typical trade-off when dealing with long time constants).

Fig. R7 | TCAD transient simulations of sweep rate dependent hysteresis loops. Mixed mode simulation results for two different configurations: **a**, the device only features an effective bulk resistance and **b**, an additional parasitic capacitance of 10 pF is connected to bulk terminal as well. Note that the dependence on sweep rate vanishes at lower R_B for the target sweep rates, but similar characteristics could be observed at higher rates.

All in all, we have provided simulation results that consistently connect the physical mechanism behind the firing and relaxation of the neuron with the observed experimental results, well supported by fundamental device physics. In the revised version of the manuscript we have:

1 – Included one sentence in the main text, which reads as follows:

“We address the time-domain component of the firing and relaxation process that enables neuron mimicking by applying constant sweep rate V_D ramps to the floating-bulk transistors and monitor the value of I_D (see Methods). The results clearly show different dependence of the firing and relaxing voltages on the sweep rate (Supplementary Fig.8-9), that becomes shallower at higher V_{G2} . This is ascribed to a lower bulk resistance that enables collection of excess carriers in the bulk of the transistor and a quick drop of electrostatic potential in the floating bulk semiconductor at lower R_B . The physics behind these dynamics are carefully verified using Technology Computer-Assisted Design (TCAD) simulations replicating the experimental conditions and showing excellent agreement with measurement results (see detailed discussion on simulation results in Supplementary Note 3).”

2 – Included the entire explanation and simulations as new Supplementary Note 3.

We thank once again the reviewer for this constructive comment, which we believe has helped a lot to enhance the quality of the manuscript.

Reviewer 1 comment #2

In the case of “floating body” conditions, when the devices behave as synapses, I cannot understand how these carriers remain in the device for such long times (as shown in Fig. 4c and Suppl. Fig 15). Notice that the effects remain even at low drain bias, when impact ionization cannot occur so that no extra carriers are

supplied. It is true that the floating body connection severely limits the collection of the carriers but there is still recombination, and recombination times are indeed much shorter than 2 hours. And there are also leakage currents through all the terminals of the device, i.e. holes can flow to the source and drain terminals since positive charges in the body forward biases these junctions. It is very difficult for me to understand how a cloud of carriers can remain in the device even when transistor bias is removed ($V_{DS}=0$). **How can a large population of holes remain in the device for such a long time (at least two hours) unless they are trapped somewhere?** If we look at other devices proposed as synapses, when they are programmed by appropriate electrical signals, there is always a long lasting internal modification leading to a nearly stable state of a certain resistance. In the case of floating gate transistors it is the trapped charge in the floating gate, in RRAM it is the formation/destruction of a ionic filament (either metal atoms or oxygen vacancies), in FeFETs the ferroelectric polarization, etc. It is very difficult for me to believe that in the MOSFETs considered in this work this internal variable is the presence/absence of a cloud of “permanent” carriers in the bulk. **This is my main objection and I would like the authors to explain this in detail.**

We deeply thank the reviewer for raising this important aspect. We completely agree with the reviewer in the fact that recombination times are indeed much shorter than 2 hours, and that excess carriers cannot explain the long-term retention of the device in the synaptic regime. Since the discussion on this mechanism was too scarce (due to the word count limits imposed by Nature), we understand why this issue arose. We hereby provide additional experiments and arguments towards the understanding of the underlying operating mechanisms of the synaptic regime in these devices, which relies on the floating bulk condition but it must relate to a trapping mechanism in the long term. For clarity, we have split these long comments into small sentences and answered them one by one.

In the introduction the authors state that: *“The connections between multiple processing units are 57 known as synapses, which need to change their electrical resistance in a non-volatile manner to favor (facilitate) or limit (depress) the connections between specific neurons”*. Thus, a defining property of a synapse is non-volatily”.

We thank the reviewer for pointing this out and we’d like to propose a revision to the text since, due to extension reasons, the interpretation of non-volatility in synapses may be oversimplified as is.

Correcting ourselves, synapses are not necessarily non-volatile. From the understanding of short-term to long-term memory conversion in biological systems, a non-volatile synapse would certainly represent long-term memory of a learnt process or task within a neural network. In many applications of artificial neural networks, this is a desirable characteristic because, in general, a specific neural network is trained (synaptic weights are set to certain values) to perform a certain function based on known input data (offline training). The learning process allows to tune the output to the desired outcome as known data is fed by updating the weight of the synaptic elements. Afterwards, the neural network is expected to maintain its operating conditions through time as external information reaches the system, in a process known as inference. In such cases, retention of the synaptic weights in non-volatile memory elements is desirable.

Even so, in many state-of-the-art implementations of in-memory computing accelerators for hardware-based artificial neural network applications rely on volatile memories such as SRAM cells^[RL6] to implement synapses, which are intrinsically volatile (the data is lost when power is removed). This is mentioned to clearly differentiate between the term non-volatility for an electronic memory device and the property of certain synapses of maintaining their strength within specific neuromorphic processes: an artificial synapse does not imply the use of a non-volatile memory device.

However, multiple neural processes require synapses to be “plastic” (i.e., synaptic plasticity) in the short-term, in the sense that they should be able to potentiate (learn) and relax (forget) the strength of a connection based on the amplitude and time-spacing of the excitation that these synapses receive at their inputs. This

process is typically known as working memory, in the sense that it is used to process certain information to complete a given task, but such information is quickly discarded upon completion (examples of such processes are speech recognition, information filter, or spatial orientation perception^[RL7]). In that sense, those synapses are indeed volatile, but still show some degree of persistence, which translates into a progressive decay of its synaptic strength with time when no excitation is present. Therefore, a synapse is not intrinsically non-volatile, even though many mainstream applications of artificial neural networks may benefit from memory devices that provide such characteristic.

In the revised version of the manuscript, we have:

1 – Included one sentence in the main text (Section “Introduction”), which reads as follows:

“ANNs are expected to outperform traditional computers in terms of energy efficiency because they can compute and store the data at the same location, which avoids energy consumption and delays related to data transfer. To do so, ANNs require: i) electronic neurons capable of generating output signals that resemble a highly nonlinear hysteretic/thresholding mathematical operation when receiving multiple voltage/current excitatory inputs^{4,5}; and ii) electronic synapses capable of changing their electrical resistance to favour (facilitate) or limit (depress) the connections between specific neurons, a process that characterizes the learning of a feature. The persistence of these changes through time (plasticity) depends on the role of the synapse within the ANN, and can be long- or short-term (see definitions in Supplementary Note 1 and Supplementary Fig.1)⁶. In several types of ANNs neural/synaptic behaviours take place dynamically (when applying electrical pulses, not continuous bias over time) which is highly desired to reduce energy consumption and provide synchronism with other parts of complex systems.”

2 – Included detailed definitions on synaptic and neural dynamics as new Supplementary Note 1.

They claim that their device can behave as synapses but afterwards they say (with respect to different stable states when under floating body conditions: *“Though this process should not be interpreted as non-volatility (from its memory conception), this synaptic state retention characteristic is certainly long-term, as the weight is maintained through more than 2 hours without requiring any type of dynamic update or refresh”*). I don’t understand the difference between “non-volatility” and long-term retention. I think that the authors should clarify this point. Why they say that their states are non-volatile but behave as synapses? Is it a question of the retention time? Why the authors mention this subject even if they don’t have any indication of memory loss during their experiments (Fig. 4c)?

We thank the reviewer for noticing this inconsistency in the text, which seems to contradict previous statements in its current form. We appreciate the chance to revise this for clarity.

As the reviewer clearly pointed out, we have no indication of memory loss during retention experiments in Fig. 4c and newly added retention experiments (following the suggestions of the reviewer in a later comment we have labelled as reviewer comment #6 further along this letter). However, from the memory conception of the term non-volatility, we prefer not to claim that the devices show non-volatile characteristics as such. It is often observed in the literature that a memory device is claimed to be non-volatile without the proper evidence to support such claim, mainly from the experimental procedure point of view^[RL8]. Retention tests of industrial non-volatile memories are typically carried out after cycling large arrays of devices for at least 10 % of their certified endurance and then tested for retention at high temperatures above 85 °C (being temperature a contributing factor to degrading retention in most memory technologies).

For the sake of clarity, we have avoided the comment on the non-volatile nature of the process. In the revised version of the manuscript, we have:

1 – Removed the phrase on non-volatility in the main text (Introduction), and included a succinct definition with a reference to new Supplementary Note 1 (which was added following this reviewer’s Comment #4 in this letter). The revised introduction now reads:

“ANNs are expected to outperform traditional computers in terms of energy efficiency because they can compute and store the data at the same location, which avoids energy consumption and delays related to data transfer. To do so, ANNs require: i) electronic neurons capable of generating output signals that resemble a highly nonlinear hysteretic/thresholding mathematical operation when receiving multiple voltage/current excitatory inputs^{4,5}; and ii) electronic synapses capable of changing their electrical resistance to favour (facilitate) or limit (depress) the connections between specific neurons, a process that characterizes the learning of a feature. The persistence of these changes through time (plasticity) depends on the role of the synapse within the ANN, and can be long- or short-term (see definitions in Supplementary Note 1 and Supplementary Fig.1)⁶. In several types of ANNs neural/synaptic behaviours take place dynamically (when applying electrical pulses, not continuous bias over time) which is highly desired to reduce energy consumption and provide synchronism with other parts of complex systems.”

2 – Removed the phrase about non-volatility in the main text and re-arranged a revised section “Long-term synaptic plasticity” to clearly address this regime in detail (further discussed in the response to the next comment inline in this letter.

3 – Briefly state the application of short-term synaptic updates in revised Section “Short-term synaptic plasticity in single floating-bulk transistors”, by including a sentence that reads:

“This regime relies on characteristic carrier lifetimes and device capacitances that translate to a precise control of conductance state and operating range, and can have immediate impact in neuromorphic computation strategies such as recurrent neural networks³³ or reservoir computing³⁴, where emerging devices (such as memristors) struggle due to inherent variability.”

4 – Briefly state the application of long-term synaptic updates in revised Section “Long-term synaptic plasticity in single floating-bulk transistors”, by including a sentence that reads:

“Next, we extend the analysis of the floating-bulk transistor for mimicking long-term synaptic plasticity. This feature is used in more widely adopted compute-in-memory accelerators based on hardware-implemented ANNs^{10,35,36}.”

Although I am not questioning the experimental results, but the physics-based interpretation of the results is extremely important. If the device behavior is due to an intrinsic effect of MOSFETs biased in an unconventional way (punchthrough/impact ionization), then, it would be feasible to properly design the devices with desired properties. If, on the contrary, the effects are related to trapping of charges somewhere in the device, this would be a serious drawback.

We totally agree with the reviewer on the fact that the displayed non-volatility of the synaptic behaviour (retention for more than 2 hours) cannot be fully explained by carrier diffusion and recombination dynamics. Instead, some degree of charge trapping should be taking place in the device as well. In fact, we discussed this very briefly in the manuscript (due to extension reasons), but we can see how this discussion can be insufficient.

We do recognize that charge trapping is, most likely, contributing to the retention of the different states. However, the reviewer states that “trapping of charges somewhere in the device would be a serious

drawback”. In this case, we consider that this effect would not necessarily represent a drawback, if harnessed properly. Mainstream embedded memories typically feature charge trapping mechanisms into floating structures for their operation. Moreover, the so-called charge-trapping transistor (CTT) has been widely proposed as a simple alternative to more elaborate embedded memory solutions for their application in storage and in neuromorphic computing applications^[RL9–RL11]. In these devices, the intrinsically “trappy” gate dielectric of HK based technologies is exploited as means of synaptic mimicking using standard technology devices. With this in mind, the existence of charge trapping processes in the floating body devices is not necessarily a drawback as long as it is repeatable within requirements: bear in mind that the 10^5 cycles of endurance shown in Fig. 4e-f is higher than qualified specifications of some commercial embedded FLASH memories for code storage^[RL12] featuring 10^4 cycles, and that charge trapping transistors have been reported for 1,000 cycles in the literature^[RL13]. From this point of view, in the case of the floating bulk device, we do not see this as a drawback but as a phenomenon that can be exploited.

Now, from the theoretical aspect, it is most likely that the injection of hot electrons during the reset (negative drain bias) process at a floating bulk condition is contributing to increase the threshold voltage (namely, HRS). Meanwhile, detrapping of some of these injected charge and/or hot hole injection (HHI) could be the mechanism through which the threshold voltage is reduced back to its initial value. In the following, we discuss these two processes (reset and set) individually, for clarity.

In the reset process (increase of threshold voltage or, in other words, increase of resistance under constant bias), a negative bias is applied to the drain at floating bulk condition. Note that, if the transistor bulk was indeed grounded, the current would be determined by the forward bias drain-bulk junction and would rise rapidly, since it would only be limited by the semiconductor spreading resistance and interconnect resistance (see measurements and TCAD simulation results for these conditions in Fig. R8a). But with the floating bulk, the decreasing drain voltage tends to forward bias the drain-body junction, lowering the electrostatic potential of the silicon bulk and inducing an inversion channel under the gate (recall that V_G is held at constant voltage). Since the source is held at 0 V, it is biased above the electrostatic potential of the body and large currents are driven in the device channel.

It is important to highlight that the observed hysteresis during dual sweeps is related to the parasitic bulk capacitance in the device and interconnects (see difference in simulations with -green dotted- and without -lilac dotted- external additional capacitance), which again can maintain the substrate potential low until it discharges through leakage sources. However, this process eventually relaxes as the total bulk capacitance discharges. This is most likely the origin of the relaxation observed in retention measurements (discussed further along this letter in the answer to reviewer comment #6).

When the drain voltage is sufficiently negative, energetic electrons are present in the vicinity of the source terminal (see impact ionization rates from the TCAD calculations in Fig. R8b) and are likely to be injected into the gate oxide. This will logically result in an increase of V_{th} and therefore a reduction of current drive capability. This tends to shift the V_{th} to a higher voltage, therefore lowering the drive current capability of the transistor, which is translated to a high resistivity state under read conditions.

(continue in next page)

Fig. R8 | Device conditions during reset sweeps. **a**, Characteristic I_D - V_D reset sweeps ($V_D < 0$ V) of the floating bulk transistor operating in synaptic regime and the same device with the grounded bulk (forward drain-bulk junction bias). The full curves are measurements and the dotted curves are TCAD measurement results (differences are expected due to lack of detailed information on the exact parameters of the front-end-of-line). Note that in all cases, the current of the floating bulk device is much smaller. Floating bulk simulations were carried out using a large resistance connected to the bulk ($10^{13} \Omega$) with a capacitor in parallel (1 fF or 10 pA). The larger capacitance at the bulk terminal explains the observed hysteresis. **b**, Electron velocity (left) and impact ionization rate (right) at the floating bulk condition when applying -3 V at the drain terminal. Note the presence of hot electrons near the source, at the oxide-silicon interface.

During the set process, some detrapping of the charge generated during the reset sweep can be expected, but the damage by hot electrons is typically non-reversible^[RL14] (at least without annealing conditions). Therefore, it is likely that the injection of holes through the gate dielectric on the drain side may also take place at impact ionization conditions. This is a well-known process that has been observed in standard silicon transistors since 1986 (Ref.^[RL15]). This phenomenon takes place at high drain voltages, where band-to-band tunnelling is likely in the drain-bulk junction and highly energetic holes and electrons are present at impact ionization conditions. Since excess holes are not collected by the bulk current in the floating bulk condition and the density of holes tends to increase at the oxide interface close to the drain (as discussed previously in Fig. R4), conditions are given for hole injection. Just like with electrons, hot holes can be injected through the gate dielectric if they have enough energy to overcome the energy barrier. As a result, this effect has been observed to be responsible for read disturb instability in EEPROM memories^[RL16,RL17] and is employed as an erase mechanism in some commercial embedded flash memories^[RL12], as injected trapped holes result in a decrease of the threshold voltage. In such cases, the hot holes that are injected can be effectively concealed within the floating gate structure, but in standard MOSFET structures a lesser number of holes can be trapped in defect centres of the gate-oxide and/or spacer oxide of the MOSFET structure.

To address the nature of the observed memory mechanism, we conducted retention measurements of the floating bulk device under pulsed regime (so as to avoid any kind of read disturb effect) at room temperature and 85 °C (see within answer to reviewer comment #6). These measurements are further discussed in response to reviewer comment #6.

We have also performed retention measurements to address the permanent charge trapping behaviour without accounting for any floating bulk effects during the retention reading period. This is done by performing set and reset sweeps (see Fig. R9a) under a floating bulk condition and then running the retention experiment with the bulk of the device grounded (see Fig. R9b), followed by I_D - V_G curves to address the V_{th} shift (ΔV_{th}) after each measurement (see Fig. R9c). These results show that a long-term synaptic effect is possible without the need for a floating gate structure by exploiting intrinsic charge trapping in standard silicon transistors aided by the floating bulk configuration. Note that different V_G voltages upon read can also enable a trade-off between readout sensitivity, power consumption and speed.

Fig. R9 | Retention measurements with grounded bulk of the synaptic device after set/reset in floating bulk condition. **a**, Characteristic I_D - V_D sweeps denoting the floating bulk transistor operation in synaptic regime. The body of the transistor is left floating (single device cell). **b**, After the set (red) and reset (yellow) sweeps, retention measurements are carried out with the bulk of the device grounded and under a periodic read regime (see insets). **c**, I_D - V_G characteristics of the device after each retention measurement from panel **b**. The shift of the curve denotes charge trapping and the impact of interface states. Note that in the 500 nm device, nominal operating voltages are never exceeded, indicating that the trapping process is not a result of high operating voltages but of the operation under floating bulk condition.

All in all, the floating bulk condition allows to use a single, standard, silicon transistor in a similar fashion as charge trapping transistors based on high-k oxides. **It is very important to highlight that in this regime the nominal operating voltage of the device (5.5 V) is never exceeded, so the displayed memory effect is not related to a high voltage regime but to the bipolar operation of the device in the floating body configuration. More importantly, the floating body enables the possibility to change between hot-electron injection and hot-hole injection without high forward bias currents (see Fig. R8a).** This enables very relevant and commercialized technologies for mixed-signal circuits (above 24 nm, without high-k dielectrics) or even enhance the trapping processes with high-k/metal gate stacks to include high density memory/synaptic devices without the need of additional process steps.

With this additional analysis, in the revised version of the manuscript, we have:

1 – Extended the discussion with a full paragraph on the role of charge trapping (Section “Long-term synaptic plasticity in single floating-bulk transistors”, paragraph 4), including references to new supplementary simulations and measurements, which now reads as follows:

“This bipolar drain bias regime cannot be employed in grounded-bulk devices, which would show high drain-bulk forward bias currents during the depression sweeps ($V_D < 0$) and no impact-ionization firing during potentiation ($V_D > 0$). In this regime, carrier lifetimes and internal device capacitances cannot be accounted for the observed results. Therefore, to explain the long-term retention we consider a contribution from carrier injection into the gate dielectric driven by impact ionization in the floating bulk device. These charge trapping processes are not a reliability concern for these devices. First, because they are operated within nominal voltages in all conditions (all voltages within 5.5 V). Second, the amount of charge trapped and de-trapped in every cycle does not trigger device failure even after 100,000 cycles (Fig.5e-f). This endurance is higher than that of some transistor-based commercial memory technologies, such as charge trapping transistors (1,000 cycles)³⁸ and embedded FLASH (10,000-100,000 cycles)³⁹, and some commercial memristor technologies (100,000 cycles)⁴⁰. The obtained results (Fig. 5e-f) for the floating-bulk device are in well agreement with this requirement. We perform TCAD simulations of the floating bulk MOSFET ($L_{CH}=500$ nm) and observe that conditions are given for hot-electron injection during depression and hot-hole injection during potentiation, mechanisms that are used to write and erase commercial floating gate memory technologies⁴¹ (see Methods, Supplementary Note 4 and Supplementary Fig.27). This is further validated experimentally by performing grounded-bulk measurements of retention and post-

retention I_D - V_G curves (see Methods), observing sustained retention and a clear subthreshold characteristic shift for each condition (see Supplementary Fig.28). ”

2 - Included the full discussion provided in this letter regarding the new retention measurements and simulation results as new Supplementary Note 4 and new Supplementary Figs. 25 and 26 (Figures R8 and R9 in this letter).

Reviewer comment #3

The authors present results from a compact model. However, assessing the physics of these devices under the reported conditions would require a physics based simulation. There are several commercial packages which allow to do these simulations in a relatively easy way. **I strongly encourage the authors to do this type of physics-based simulations.** In particular, this might allow to unveil the microscopic nature of the experimentally observed behaviors and to respond to my main objection related to synaptic behavior and charge accumulation.

We thank the reviewer for their suggestion. The objective of the SPICE compact model is to obtain a simulation-compatible representation of the device’s physical behaviour (under the neural regime) that is well validated from the physical principles governing the components being used within the model. Please note that no ideal behavioural elements (like current or voltage-controlled sources of any kind nor algebraic loops within the model) were used to obtain a first order representation of the device characteristics.

That being said, we understand the request for physics-based simulations that can support the electrical characterization results that take over most of the manuscript, and we appreciate this suggestion as simulation results have provided a more complete framework to the manuscript. Therefore, and as shown in previous replies to this reviewer’s comments, we have conducted finite element method simulations using Synopsys’ Sentaurus TCAD package over a 2D structure of the MOSFET device. Since certain physical parameters are proprietary to the manufacturer of the devices, some structure details are expected to differ from the actual characteristics of the measured devices, such as exact doping profiles, oxide and poly-silicon thicknesses, contact resistances, among others. Even despite these uncertainties, transient simulation results of the device show great agreement with experimental results and have been very useful towards theoretically confirming the origin of the observed behaviours. Simulation results have been employed throughout this letter to address the comments by the reviewer and have been included as part of revised manuscript:

- 1- We introduced a new Supplementary Note 3 (and Supplementary Figs. 10-14 therein), with detailed simulations of the neuron dynamics that were experimentally addressed.
- 2- We introduced a new Supplementary Note 4 (and Supplementary Fig. 27 therein), with detailed simulations of the device under fully floating gate conditions for synaptic mimicking.
- 3- We introduced a new Supplementary Note 8 (and Supplementary Figs. 35-36 therein), with details regarding simulation approach, models and description of the publicly shared code for the simulations. A revised methods section refers the reader to the publicly available code and to this supplementary note:

“Physical simulations: Physical simulations of the floating bulk transistors are carried out using commercial Technology Computer-Assisted Design (TCAD) software (Synopsys’ Sentaurus TCAD). Device 2D structure is built using a structure descriptive approach and based on commonly known parameters of the technology node at which the devices under test were fabricated. Next, the device structure is optimised by simulation of the nominal (grounded bulk) quasistationary I_D - V_G characteristics and calibration against experimental data. Once good agreement is reached, transient

I_D-V_G simulations including impact ionization physics are run to calibrate the impact ionization model parameters to the nominal substrate currents in the device. With this calibration fixed, we ran transient simulations of I_D-V_D at different V_D sweep rates within a mixed mode environment that connects the bulk biasing network to the substrate contact of the 2-D device structure. The bulk biasing network involves a constant value resistance (R_{sub}), or an N-channel transistor modelled in SPICE BSIM and biased by an independent gate voltage source. In all cases, a bulk connected capacitance (C_{bulk}) is included to account for experimental and device connection parasitics. Complete Sentaurus Workbench project including command and parameter files are available through a public repository at <https://doi.org/10.5281/zenodo.13843362>. See more details on the models used and the project structure in Supplementary Note 8.

- 4- We revised the main text to refer to the simulation results describing the mechanism, in the revised section “Punch-through impact ionization in CMOS transistors”. Last paragraph now reads:

“We address the time-domain component of the firing and relaxation process that enables neuron mimicking by applying constant sweep rate V_D ramps to the floating-bulk transistors and monitor the value of I_D (see Methods). The results clearly show different dependence of the firing and relaxing voltages on the sweep rate (Supplementary Figs.8-9), that becomes shallower at higher V_{G2}. This is ascribed to a lower bulk resistance that enables collection of excess carriers in the bulk of the transistor and a quick drop of electrostatic potential in the floating bulk semiconductor at lower R_B. The physics behind these dynamics are carefully verified using Technology Computer-Assisted Design (TCAD) simulations replicating the experimental conditions and showing excellent agreement with measurement results (see detailed discussion on simulation results in Supplementary Note 3).”

- 5- We revised the main text to refer to the simulation results describing conditions under which the synaptic behaviour is obtained, in the revised section “Long-term synaptic plasticity”. A new sentence in the last paragraph now reads:

“We perform TCAD simulations of the floating bulk MOSFET (L_{CH}=500 nm) and observe that conditions are given for hot-electron injection during depression and hot-hole injection during potentiation, mechanisms that are used to write and erase commercial floating gate memory technologies⁴¹ (see Methods, Supplementary Note 4 and Supplementary Fig.27).”

Other comments and suggestions

Reviewer 1 comment #4

I suggest that, in the interest of the general reader it would be convenient to introduce a supplementary note explaining the details of what is the difference between (a) integrate-and-fire neural behavior, (b) tunable neuro-synaptic plasticity and (c) synaptic behavior.

We thank the reviewer for this suggestion, and we totally agree on the importance of discussing the main concepts behind the neural and synaptic behaviours that are covered in our work. We have included a new Supplementary Note 1 that briefly introduces the readers to the fundamentals of neuromorphic behaviours.

Reviewer 1 comment #5

In the case of memristive devices, the origin of cycle-to-cycle (C2C) variability can be understood at least qualitatively (changes in the filament in RRAM, changes in the melted volume in PCRAM, etc.). However,

it is harder for me to understand C2C variability in a well-control device such as a MOSFET based on deterministic processes (unless charge trapping were involved). What is the physics-based origin of the cycle-to-cycle variability (both the ON and OFF resistances) as shown in Fig. 3a and suppl. Figs. 9 and 10? Can the authors comment on this?

We thank the reviewer for noticing this. Multiple experiments carried out in this work are long-term, which means the full measurement may last between a few minutes to up to 1 day. The cycle-to-cycle variability can be therefore explained in two parts: the short-term variation and the long-term variation.

On one hand, the short-term variation is mainly dominated by instrument noise under time-domain pulsed/high bandwidth measurement conditions and at very low currents. Devices 1 to 9 were not connected to an external transistor through the chuck, but instead a constant resistance was connected to ($\sim 1 \text{ K}\Omega$) in addition to the substrate spreading resistance of a $5 \text{ mm} \times 5 \text{ mm}$ die (which can be in the order of $1 \text{ M}\Omega$). In such conditions, the floating bulk transistor is the neuron regime (these details have been included in the caption of revised Supplementary Fig. 15 for completeness). Note that in Supplementary Fig. 9 of the original version of the manuscript (now Supplementary Fig. 15), the waveform generator/fast measurement unit (WGFMU) used to measure the current through the devices is set to a range of $10 \mu\text{A}$ (e.g., device 1) or $100 \mu\text{A}$ (e.g., device 7) to accommodate the firing currents. Additionally, only 5 points in the full forward+backward sweep are stored in memory, to reduce data transfer in the instrument and make the experiment feasible in terms of execution time. Under such conditions, the low current level (high resistance) before firing is mostly dominated by instrument noise, which is higher close to the resolution limit (which shows as a wide band of discrete values for device 1). Given that the acquisition channels are linear, only a fraction of the resistance jump can be captured under continuous ramping used to induce the fire and relax within experimentally feasible times (full experiment lasting < 10 hours). This means that instrument noise can also impact on the repeatability of the measurement at the high-current (low resistance) state of the device, as can be clearly observed from the wider spread of resistance for device 1, which are measured farther away from the top of the range, where the relative impact of noise is lower (see Fig. R10). This doesn't affect the interpretation of the results, since the device is clearly firing and relaxing between cycles in all cases, and the effective resistance measured is merely an expected result of the experimental measurement conditions.

Devices 8 and 9 (in revised Supplementary Fig. 15) correspond to a different die from devices 1-7, showing different spreading resistance of the substrate and therefore different firing conditions and noise in the measurement (as was displayed in Supplementary Fig. 5 of the revised manuscript). Still, the source of the variability is inherent to the measurement conditions and not to the operating mechanism of the floating body device. For this reason, as discussed in Supplementary Figs. 5 and 7 of the revised version of the manuscript, the bulk control transistor helps better fix the firing conditions by well establishing the impedance of the bulk connection, but adding parasitic capacitance to the measurement setup in the process.

(continue in next page)

Fig. R10 | Measurement setup sensitivity and noise. Examples of device 1 and 7 for the impact of noise in the acquisitions at different measurement ranges (sensitivities of the measurement channel). Note that the noise is always larger at high resistances and when measuring farther away from the maximum measurement range of the setting. This gives rise to measurement variability that is not related to the device itself.

On the other hand, the long-term variation can be observed throughout the capture of millions of cycles. In all the experiments, best efforts are invested on keeping ambient conditions under control: vibration isolation tables, enclosed measurement chambers, uninterrupted power source unit to regulate line voltage variability, temperature- and humidity-controlled environment, among others, are all in place as part of accepted requirements from a semiconductor electrical characterization testbench. Even under these conditions long term drift and stability are always present, affecting mostly tip-to-pad contact stability and repeatability (between devices), die-to-chuck effective resistance, chuck noise and slow thermal drifts of sample, test fixtures (probes, probe arms, chuck, station) and equipment throughout the day, in addition to equipment long-term drift between calibrations. These sources of variability can be clearly identified in the measurements (see Fig. R11), observed as a correlation between the high and low resistance levels displayed in the scatter plots. However, once again, they do not affect the interpretation of the results.

Fig. R11 | Connection instability in long term measurements. Examples of device 8 and 9 for tip to pad instability giving origin to discontinuities in the measurement that are not related to the device itself.

Therefore, the origin of the observed variability is not related to the device itself, but with experimental conditions of this test. An alternative setup to address the firing is the use of the neuron in burst firing mode by introducing a current excitation and monitoring the voltage through a high-input impedance amplifier (see Fig. R12). Under these conditions it is possible to see that the firing characteristics are consistent through multiple cycles, further highlighting the origin of the instabilities observed in endurance measurements is not related to the device itself but to the measurement setup conditions.

Fig. R12 | Repeatability of the firing characteristics of the floating body neuron in bursting mode. a, The floating bulk transistor is connected in the two-device configuration to a current source, a voltage amplifier and an oscilloscope through a high-input impedance amplifier. **b,** Relative firing amplitude across 150 cycles. Note that the observed variability is very low and only related to +/- 1 count of the oscilloscope's voltage channel resolution. **c,** The repeatable periodic firing under $V_G = 0.45$ V, $V_{G2} = 1.8$ V and excitatory input current $I_{excitatory} = 30$ μ A.

With the origin of variability clearly identified for our experimental conditions, we would like to highlight that it is always our policy to show the characterization results “as captured”, even though noise or some expected degree of measurement instability can result in less “clean looking” data plots. As long as these sources of variability do not affect the interpretation of the results, we firmly believe that experiment non-idealities should be clearly displayed, within reasonable limits, for reproducibility of the experiments and results.

In the revised version of the manuscript, the caption of now Supplementary Fig. 15 has been revised to include this clarification on the origin of variability. The revised caption now reads:

“Observed cycle-to-cycle variability is acquisition noise of the time domain measurement and not related to the device itself. In all cases, V_G is biased between 0.36 V to 0.45 V, peak V_{spike} applied to V_D between 4 V and 4.4 V. Devices 1-7 and 8-9 correspond to two different die from Supplementary Fig. 5. Bulk terminals were not connected to an external transistor through the chuck, but instead a constant resistance was connected (~ 1 K Ω) in addition to the substrate spreading resistance of a 5 mm \times 5 mm die (estimated 0.1-1 M Ω). Devices 10-12 correspond to same die as 8-9, but connected to a bulk control transistor (as in Supplementary Fig. 7) with $V_{G2} = 1.6$ V.”

Reviewer 1 comment #6

The conditions under which retention measurements are performed (Fig. 4c, suppl. Fig. 15) should be specified (V_G, V_{DS}). What would happen if V_{DS} were kept at zero volts and retention measured by successive pulses? Would the device relax to “equilibrium” during the $V_{DS}=0$ periods? I suggest to do some measurements in this regard.

We thank the reviewer for this suggestion. We also thank them for noticing that we missed the bias conditions of the device in Fig. 4c, Supp. Fig. 15. We have included these details in the revised captions of each figure, now stating that $V_G = 0.7$ V and $V_{DS} = 0.5$ V, in all cases.

Regarding retention chronoamperometry measurements, we used a relatively low $V_{DS} = 0.5$ V (compared to the characterized DC voltages that induce state changes) to ensure that there is very little influence from the applied bias on the effective state of the device throughout the measurement (also known as read disturb). However, we agree with the reviewer that removing the read voltage between read events more closely represents the working conditions for a device being tested for retention. For this reason, we conducted new parametric experiments that address the retention characteristic of the floating body transistors in synaptic regime, this time by applying 100 ms read pulses of 0.5 V every ~1s, during a lapse of >3 hours (see Fig. R13). These new sets of measurements were conducted at room temperature and at 85 °C. Even under these conditions, the observed state is persistent through more than 3 hours, in coincidence with our previous experiment. At high temperatures, measured resistance is smaller, as expected, due to the increase in overall leakage and subthreshold currents in the MOSFET device. Although the memory window becomes narrower at high temperatures, at least 4 stable levels can be clearly distinguished throughout the whole measurement.

(Continue in next page)

Fig. R13 | Retention measurements under long-term periodic read. **a**, I-V characteristic of reset sweeps (up to different reset voltages ranging from -2.25 V to -4 V) and resulting read I-V sweeps up to 0.5 V (performed after >3 hour retention measurements from panel b). The inset shows the post-retention read I-V sweeps in linear scale. Every reset sweep is preceded by a SET sweep up to 5 V. **b**, Retention results measured after each reset sweep. The read voltage is held at 0 V between read pulses lasting 100 ms, with a period of ~1 s. In both panels, $V_G = 0.5$ V. The long-term retention is preceded by a short-term transition. Each regime can be exploited in different synaptic implementations. **c**, Same measurements as in panel **a** but at 85 °C. **d**, Same measurements as in panel **b** but at 85 °C.

Lastly, a comment is worthy on the observed short-term to long-term transition, particularly in measurements performed at room temperature (see Fig. R13b). This effect could be already observed in previous measurements conducted at constant read voltage (chronoamperometry) tests from Fig. 4c of the previous version of the manuscript. To account for this, it is important to consider the previous history of the device for each measurement. In the lowest resistance state reported, it is possible to see a transition towards 10 M Ω , where the device stabilizes (see black curve in Fig. R13b). In this case, the retention measurement is conducted right after a set/potential sweep that drove the floating bulk device into the avalanche region. In the rest of the measurements (colour lines), the retention is analysed after a reset/depression sweep. The impact of substrate capacitance has been discussed in the reply to reviewer comment #2. This capacitance maintains the substrate biased at a lower potential than the source, deepening the depletion region and increase device threshold voltage, which translates into a higher channel resistance during the first section of the retention measurement. Following this slow transition lasting in the range of 100~1000 s, the current stabilizes at well-defined levels, related to the charge trapping effect (discussed in Fig. R9). This dynamic can be useful for the application of the synaptic device under different plasticity regimes that include both a short-term inhibitory response and a long-term depression response related to permanent charge trapping, with a slow transition between both regimes. However, this term will be largely dependent on overall substrate connection capacitance.

To highlight these new results, in the revised manuscript we have:

- 1- Introduced changes to the second paragraph in revised section “Long-term synaptic plasticity”, which now reads:

“We also assess the retention at constant read voltage of at least 6 resistance states after different reset voltages (see Methods), with long-term synaptic stability through 10^4 seconds (~2.8 hours, Fig.5c) without any kind of refresh. This behaviour still holds when the readout consist of a train of 100-ms-long pulses spaced over 900 ms (see Methods and Supplementary Fig.25a,b) — discarding that the state would be maintained artificially by the read voltage applied — and even when tested at high temperatures (85 °C, see Supplementary Fig.25c,d). We also note that this process can be pulse driven, initially with pulses in the range of 10~100 ms (see Supplementary Fig.26). Therefore, to address the repeatability of this process under a pulse regime that better suits real application conditions, we evaluate the switching performance using 500 μ s pulses of voltages +5 V for potentiation, -3 V for depression, 0.35 V for reading the acquired state after each pulse (Fig.5d), observing that the devices consistently switch between two distinctive resistance states (ratio > 10 \times) for more than 10^5 cycles (Fig.5e and 5f).”

- 2- Included the new measurements from Fig. R13 as revised Supplementary Fig. 25.

- 3- Revised the Methods section clearly indicating the conditions under which the retention measurements are carried out. The section *Device basic characterization* now includes the following additional text:

“Retention and pulsed-read retention measurements (Supplementary Figs.25, 26 and 28) are carried out in Keysight’s EasyEXPERT environment using a custom arrangement of I-V, I-V pulse and I-V list sweeps. In the floating/non-floating bulk retention tests (Supplementary Figs.26) that address the permanent nature of the device state, the substrate tap contact is opened using the built-in SMU switch for the floating bulk writing procedures and then grounded through the SMU during the retention period.”

Reviewer 1 comment #7

Some minor corrections

Page 5, line 266: *we operate 500 nm thick-oxide transistors in the fully floating body condition (V_{G2}) with a bipolar voltage scheme.* Shouldn’t it be $V_{G2}=0$?

We thank the reviewer for noticing this. During these experiments, the bulk control device (biased through V_{G2}) is disconnected, which means the main device operates at fully floating bulk condition. In the revised version of the manuscript (Section “Long-term synaptic plasticity”, paragraph 1) we have revised this sentence for clarity:

“We now tackle the potential of the floating-bulk n-type MOSFET to operate as synapse. To mimic this behaviour, we operate individual transistors ($L_{CH}=500$ nm) with a bipolar voltage scheme in the fully floating-bulk condition (bulk bias control transistor is not connected, and the bulk terminal is left unbiased).”

The description of Fig5(b) seems incomplete ;what is the difference between the left and the right curves (resistance versus CDF)? By the way, more standard plots would show CDF versus the independent variable (R in this case) and not vice versa.

We appreciate the reviewer for highlighting this clarity issue. The left and right Resistance vs CDF plots in Fig. 5b represent the empirical cumulative distribution functions of each level for the potentiation and depression cycles, respectively. To reflect this, the background colour of the axis was used to match those of the centre plot, but we understand this may not be clear for the reader. We have therefore revised Fig. 5b and included labels to clearly indicate potentiation and depression in each of the CDF plots.

Regarding how typical CDF plots are displayed, we are aware that we have alternated the axis in our representation. The reason is to match the orientation of the typical multilevel synaptic plot showing resistance levels as a function of the applied input spike/pulse (centre plot). This way, in addition to the statistics of the bar plot, the readers can directly observe the full distribution of the levels on the CDF in the same orientation. Though we agree this is not typical, we humbly believe it doesn't affect the interpretation of the data and it provides a more direct visual interpretation of the statistics.

In the revised version of the manuscript, we have clarified this in the revised caption of now Fig. 4 (note that Fig. 4 and 5 have been swapped to improve the flow of the text following editorial suggestions) and included labels on the axis of now Fig. 4b to clearly indicate which data are being represented in each case.

The explanation of the sweep rate dependence of the “threshold voltage (V_{DS})” seems reasonable in the main text page 3, line 160-162. However, the explanation in the figure caption of Suppl. Fig 4a/b is in contradiction with it.

We thank the reviewer for pointing this out. This apparent contradiction may arise from the result displayed for 1000 V/s, where the threshold voltage seems to be lower than for 100 V/s sweep. This is an artifact of the measurement conditions: at 1000 V/s, the sweep rate is faster than the firing, causing the ramp to be in its back-sweep at the moment the device is fired. For clarity, we have performed new experiments assessing the sweep rate influence on the hysteretic characteristic, as already discussed in the reply to reviewer comment #1.

In the revised version of the manuscript, we:

1 - Provide revised Supplementary Figs. 8 and 9 (featuring data included in Figs. R5 and R6 from this letter) alongside a new Supplementary Note 3, introducing new TCAD simulations of the sweep rate dependence and new detailed measurements of these dynamics.

2 - Revised the main text in page 2, last paragraph referring to these new measurements. The paragraph now reads:

“We address the time-domain component of the firing and relaxation process that enables neuron mimicking by applying constant sweep rate V_D ramps to the floating-bulk transistors and monitor the value of I_D (see Methods). The results clearly show different dependence of the firing and relaxing voltages on the sweep rate (Supplementary Figs.8-9), that becomes shallower at higher V_{G2} . This is ascribed to a lower bulk resistance that enables collection of excess carriers in the bulk of the transistor and a quick drop of electrostatic potential in the floating bulk semiconductor at lower R_B . The physics behind these dynamics are carefully verified using Technology Computer-Assisted Design (TCAD) simulations replicating the experimental conditions and showing excellent agreement with measurement results (see detailed discussion on simulation results in Supplementary Note 3).”

The pulse amplitude for depression and the reading voltage are different in Fig 4(b) and in the description of the same figure in the main texts (page 6 line 285).

We deeply thank the reviewer for noticing this error. The correct value is 0.5 V and text has been revised accordingly.

The values of potentiation and depression voltages in Suppl. Fig. 16 do not coincide with those reported in the text (Page 6 line 304)

We thank the reviewer for noticing this. We have revised the text and figure captions to clearly indicate the pulse parameters in each of the experiments in the panels from Fig. 5, and Supplementary Figs. (which have been renumbered 27 and 28 in the new version).

In Suppl. Fig 11, the bulk gate voltage is named “ V_{Gb} ”. In the text, it is V_{G2} . This is the same in Fig. 3d and 3e (in the caption it is ok)

We thank the reviewer for finding this labelling error that resulted from nomenclature changes that were made when versioning the manuscript. All labels have been replaced with V_{G2} both in the revised main text and the new supplementary information file, to maintain consistency.

Referee #2 (Remarks to the Author):

This paper investigates using standard silicon CMOS transistors to replicate electronic neural and synaptic behaviors when what the authors call biased unconventionally. The authors propose this method as the feasibility of a potential short-term solution for implementing neuromorphic circuits in artificial intelligence systems while considering the challenges associated with traditional MOSFET and CMOS-based approaches. While the authors have provided lots of experimental results from two device types (nMOS with two different oxide thicknesses), the results are not fully surprising as the kink effect from Figure 1b is also observed when the IDVG of FDSOI devices is investigated [1]. The kink the authors see in Figure 1b originates from the impact ionization effect, which the authors explain. However, the bulk must remain floating to see such a kink, similar to FDSOI. By adding a transistor to the bulk contact to mimic a series resistor with variable resistance – from high R when V_{G2} is 0V to low R when V_{GS2} is above the V_{th} of the bulk transistor – the authors can switch between an FDSOI device and a conventional nMOS transistor, where the bulk is typically grounded.

We thank the reviewer for reading our manuscript and providing his/her insight. We agree with the reviewer on the fact that the kink effect was discovered years ago in FDSOI.

What is completely unknown for the community is that this effect in MOSFETs can be exploited to mimic multiple neural, long-term synaptic and short-term synaptic behaviours with excellent performance, energy efficiency, low variability and, at the same time high tunability and low area overhead. The 2-transistor-cell capable to operate in all different regimes offers unique versatility to build ANNs never reported before. Therefore, we think our results are very surprising: the MOSFET transistor has been used for 75 years and nobody achieved such kind of performance, which is very promising for ANN implementation.

We humbly believe that employing a well-known physical mechanism, in a widely proven, mature technology to solve a pressing state-of-the-art problem (implementation of neural and synaptic behaviours) is definitely an advantage compared to emerging solutions that may still require significant investment and suffer from limited performance and reliability.

Regarding the device, the authors mention that the channel length is 180nm and 500nm. Still, it would also be interesting to know which oxide thickness has been used for the investigated technology. As V_D increases to 3V and higher, I assume an EOT exceeding 7nm. The oxide thickness is an important parameter here as the show effect relies on impact ionization, and here, the oxide has to sustain the high VGD. Consequently, the synopsis behavior will only be achievable for a technology with a minimum EOT, which I assume exceeds the values of highly scaled nodes. This might be a showstopper for the investigated effect.

We thank the reviewer for pointing out this aspect. From cross-section transmission electron microscopy data, we can address that the estimated oxide thickness in 180 nm devices is ~3.5 nm and for 500 nm devices is ~10 nm (see Fig. R14a and R14b, respectively). We show these images in this response letter to answer the reviewer's comment, but they cannot be included in the revised Supplementary Information file because they contain manufacturer proprietary information.

(Continue in next page)

[REDACTED]

We agree that the operating voltages are high in some specific conditions for the 180 nm devices and that reliability is an important aspect to be addressed whenever a device is proposed to operate within specific conditions that may fall outside the nominal. However, we have to respectfully disagree on this being a showstopper, as mentioned by the reviewer, and our position can be supported on multiple aspects that we detail below:

- 1) **Operating principle can take place within nominal voltages.** In the experiments carried out in 500 nm devices, the applied voltages in our experiments never exceed nominal operating voltages (5.5 V nominal from the manufacturer) and the punch-through impact ionization process that fires the device takes place at voltages well below the nominal in the floating bulk condition (down to ~ 2.5 V). This indicates that different front-end-of-line design can be employed to facilitate this regime in case a lower operating voltage was desired.
- 2) **Tuneability allows to operate within maximum voltages.** In the 180 nm MOSFET, the neuro-synaptic regime can be achieved at lower voltages, at the cost of higher power consumption (higher currents), by modulating the neuron response through gate bias voltage V_G and substrate resistance control (V_{G2}). Under certain conditions, the firing voltage can be as low as 2 V, which is within the corner specification of the nominal voltage ($1.1 \times V_{DD}$). From the purely experimental perspective, during the experiments carried out for this work we observed only 3 events of dielectric breakdown within more than 100 devices (180 nm channel) that were intensively measured. We cannot ascribe these events to the specific voltages that were used. In fact, we have measured multiple devices under sweeping, pulsing and (newly added) bursting conditions for millions of cycles and did not notice a degradation on the behaviour of such devices. This doesn't mean to imply that these voltages would pass a very stringent, industrial reliability requirement. But, while this aspect is of utmost importance, it requires a specific qualification procedure that should be addressed in a technology/process-specific manner.
- 3) **High drain voltage by itself may not represent a risk on reliability.** Stating that the voltage is higher than the nominal is not enough to indicate a definite reliability limitation. Specific circuit configurations can drive drain voltage over the nominal in the off condition for transient periods of time (e.g., class AB/C amplifiers, see Ref.^[RL18]), and careful assessment of the reliability can show that this condition is acceptable for certain applications. For the working conditions of the devices under test, only the drain voltage is pushed to voltages higher than the nominal, while the channel remains

between the OFF and the weak inversion conditions, well below the threshold voltage. As the reviewer has well pointed out, this puts the device under high V_{DG} stress, which is at most $\sim 1.8\times$ the nominal voltage (1.5 V for this technology by this manufacturer). However, as mentioned previously, this operating voltage can be reduced to $1.1\times$ the nominal voltage and it is well known that, under OFF condition stress, the dielectric breakdown in MOSFETs takes place on the spacer side, with a long percolation path and a resulting higher voltage tolerance. Note that, for transistors of the 130nm node, accelerated time dependent dielectric breakdown experiments at room temperature are conducted at voltages as high as 16 V (namely, $\sim 10\times$ higher than the nominal voltage)^[RL19]. Therefore, a detailed reliability study should contemplate the full range of operating conditions of the device, which may limit the full operational voltage space here described to ensure reliability.

- 4) **Device front-end-of-line can be optimized for operating in this specific regime at lower voltages.** The parasitic mechanism of punch-through impact ionization being employed is an unwanted effect in standard CMOS circuits, and therefore CMOS devices themselves are designed to minimize the presence of this effect under nominal conditions, as was discussed in the original version of the manuscript. Even under these conditions, the avalanche behaviour is well-known and used in VLSI circuits as electrostatic discharge (ESD) protection devices on input/output pins, as discussed in revised Supplementary Note 2. Therefore, it is expected that higher voltages than nominal are required to exploit this effect on standard CMOS devices but is still featured in non-core devices as a standard phenomenon. Even in these conditions, for the sake of exploiting this phenomenon for this specific application, device engineering is definitely possible within standard procedures in commercial foundries, as evidenced by the wide availability of speciality devices (commercial examples include processes by XFAB, Tower Semiconductors, TSMC, among others) in current CMOS technologies that are very mature and are still widely commercialized nowadays (14 nm and above). Therefore, even if some degree of redesign and optimization of the devices is required to evacuate any possible reliability concerns, this does not diminish the potential of the proposed approach. Moreover, this device optimization would be done on a vastly well-known platform, with decades of experience on the making and with thousands of specialty devices that have been tailored to perform specific tasks in analogue/mixed-signal integrated circuits, and for this reason we see this as extremely feasible for the dominating silicon processing industry.

To clearly highlight these aspects, we have revised the main text and supplementary information including these discussions:

- 1- Added a sentence in the first paragraph of the Discussion section, highlighting that nominal voltages in 500 nm transistors are never exceeded. This paragraph now reads:

“It could be argued that the operating voltages (between 3.6 V and 4.5 V in the μ s-pulsed regime) are a downside of this device — compared to standard-operated CMOS transistors, although they are very competitive compared to memristors and other emerging devices, see Supplementary Table 1. However, punch-through avalanche is a parasitic effect under nominal operating conditions, which is minimized via specific device engineering (e.g., halo or punch-through stop implants^{42,43} or layers⁴⁴, depending on the technology). Therefore, there are multiple device-level parameters that can be engineered to enhance specific performance metrics (including operating voltages). As evidence for this, it is worth pointing out that all the neuro-synaptic behaviours demonstrated in MOSFETs with $L_{CH}=180$ nm have been reproduced in MOSFETs with $L_{CH}=500$ nm — and applying voltages within their nominal (5.5 V) — meaning that they are driven by the same fundamental phenomenon. Therefore, this regime of operation does not imply a risk to reliability (see detailed discussion in Supplementary Note 5).”

- 2- Included a new paragraph with a specific discussion on reliability aspects in the revised section titled “Long-term synaptic plasticity”. This paragraph reads:

“This bipolar drain bias regime cannot be employed in grounded-bulk devices, which would show high drain-bulk forward bias currents during the depression sweeps ($V_D < 0$) and no impact-ionization firing during potentiation ($V_D > 0$ V). In this regime, carrier lifetimes and internal device capacitances cannot be accounted for the observed results. Therefore, to explain the long-term retention we consider a contribution from carrier injection into the gate dielectric driven by impact ionization in the floating bulk device. These charge trapping processes are not a reliability concern for these devices. First, because they are operated within nominal voltages in all conditions (all voltages within 5.5 V). Second, the amount of charge trapped and de-trapped in every cycle does not trigger device failure even after 100,000 cycles (Fig. 5e-f). This endurance is higher than that of some transistor-based commercial memory technologies, such as charge trapping transistors (1,000 cycles)³⁸ and embedded FLASH (10,000-100,000 cycles)³⁹, and some commercial memristor technologies (100,000 cycles)⁴⁰. The obtained results (Fig. 5e-f) for the floating-bulk device are in well agreement with this requirement. We perform TCAD simulations of the floating bulk MOSFET ($L_{CH}=500$ nm) and observe that conditions are given for hot-electron injection during depression and hot-hole injection during potentiation, mechanisms that are used to write and erase commercial floating gate memory technologies⁴¹ (see Methods, Supplementary Note 4 and Supplementary Fig. 27). This is further validated experimentally by performing grounded-bulk measurements of retention and post-retention I_D - V_G curves (see Methods), observing sustained retention and a clear subthreshold characteristic shift for each condition (see Supplementary Fig. 28).”

- 3- Included the discussion provided in this letter as part of the supplementary information, in a new Supplementary Note 5.

Another issue I see in this work (an issue for high novelty requested by Nature Journal) is that the synopsis behavior is instead a binary switching between LRS and HRS rather than an analog synopsis. This leads to limitations in the precision of synaptic weight representations, and neural networks from binary synopsis might need help to capture complex patterns. Also, other simple and established circuits are available to mimic binary synopsis, e.g., SRAM cells, which can be scaled very well and operate the transistors within VDD, thus ensuring a more extended device and circuit lifetime.

We thank the reviewer for raising this question. We apologize to the reviewer for not being sure what he/she means by “novelty” ascribed to the binary or multilevel nature of a synapse. We agree on the fact that recent literature highlights the possibility of arbitrary conductance states and the benefits this implies for fully analogue computing, for example using oxide memristor technology^[RL20,RL21]. However, fully analogue artificial neural networks face variability and noise challenges to which digital counterparts are resilient to, which limit the number of conductance levels at which an artificial synapse can effectively operate within a neural network design. Therefore, while important and potentially useful, the analogue nature of the synaptic element does not imply novelty by itself. Instead, the novelty in our proposal is the use of a standard device to mimic the most important building blocks in artificial neural networks (neurons and synapses), avoiding the use extra components in high density neuromorphic circuits in silicon.

Moreover, we are not sure why the reviewer indicates that the synaptic behaviour in our floating bulk transistor is binary, as in the original version of the manuscript we presented at least 4 levels of configurable synaptic states in the long term (see Fig. 4c), and a deep characterization of the multi-level short-term synaptic states (see Fig. 5).

Even in the case of the synapse not being analogue-capable, the reviewer points out that this “leads to limitations in precision and implementation of neural networks”. However, some of the state-of-the-art in-memory computing accelerators for neuromorphic applications are indeed based on mainstream SRAM cells^[RL6]. These implementations are showing great potential for AI accelerators, despite some drawbacks

such as energy consumption and volatility. Therefore, the implementation of binary synapses is not necessarily a limitation for the development of artificial neural network accelerators.

The operation within the nominal VDD also enables the SRAM cell to be scaled; however, as the approach shown in this work requires unconventional high biases, the scaling options might be limited. Thus, these devices might need more chip area.

We thank the reviewer for providing their point of view. We would like to highlight that, as previously discussed, in the floating body transistors of 500 nm channel length mimicking synaptic behaviours the applied voltages never exceed the nominal voltage for these devices (foundry specification of 5.5 V nominal). Therefore, no “high biases” are used under these operating conditions. The only unconventional aspect is operating the device at a floating bulk condition to enable impact-ionization avalanche conditions during the set and reset procedures.

If, instead, the reviewer wanted to highlight the voltages employed for the neural behaviour observed in 180 nm transistors, some set of results shown in our manuscript indeed use voltages higher than nominal, but the implications at the technology level and reliability level have been discussed in an answer to a previous comment from this reviewer.

Although the work has been carefully carried out, the main focus was on describing a phenomenon of a single silicon transistor. As Si devices are pretty mainstream, I expect the authors to use the proposed effect to demonstrate a neural network on a chip.

We thank the reviewer for this comment. Different kinds of ANNs have been proposed based on different device structures and different physical mechanisms employed to implement synapses and neurons. In all cases, for a given switching mechanism, the time between its first use as neuron/synapse and the implementation of a hardware-based ANN was of at least 7 years, and not in all cases these demonstrations have been fully on-a-chip. See Table R1:

(Continue in next page)

Table R1. Technologies used for neuron and synaptic mimicking. The second column states the first report of the fundamental physical phenomenon that operates the device. The third column refers to the first experimental demonstration of the capabilities of the device to mimic neurons and/or synapses. The fourth column states the first demonstration of a full neural network, where some of them involve combination of software and hardware (e.g., the synaptic array is generally implemented in hardware and the rest of the neural network is either ran in external hardware or executed in software in a PC). The fifth column highlights the most recent, state-of-the-art neural network implemented with each technology.

Technology	Fundamental phenomenon reported	First report of neuro-synaptic behaviour	First full neural network demonstration	State-of-the-art neural network demonstration	Time from device to system level demonstration
Phase change (PCM)	Ovshinsky (1968) Ref. [RL22]	Kuzum (2011), Suri (2011) (synapse) Refs. [RL23,RL24] Tuma (2016), Sebastian (2017) (Neuron and synapse) Refs. [RL25,RL26]	Ambrogio (2018) (mixed hardware-software) Ref. [RL27]	LeGallo (2023) Ref. [RL28]	7 to 12 years (depending on what is considered a full neural network)
Metal-oxide memristors (RRAM)	Hickmott (1962) Ref. [RL29]	Snider (2008) Ref. [RL30] (synapse spiking time dependent plasticity)	Prezioso (2015) (passive crossbar vector-matrix multiply) Ref. [RL31]	Song (2024), Rao (2023) Refs. [RL20,RL21]	7 years
Ferroelectric FET (FeFET)	Moll (1963) Ref. [RL32] Ishiwara (1993) (First concept of neuron + synapse) Ref. [RL33]	Mulaosmanovic (2017), Jerry (2017) (synapse) Refs. [RL34,RL35] Wang (2018) (neuron) Ref. [RL36]	Simulations only	Kim (2023) Soliman (2023) (Simulations only) Ref. [RL37,RL38]	Pending (> 7 years)
Magneto-resistive (MRAM)	Berger (1984) Ref. [RL39]	Vincent (2015) (synapse) Ref. [RL40]	Jung (2022) (vector-matrix multiply) Ref. [RL41]	Song (2023) Ref. [RL42]	8 years
Silicon nanowires		Han (2020) (LIF neuron) Ref. [RL43]	-	Han (2022) (simulations only) Ref. [RL43]	Pending
Punch-through impact ionization / BTBT MOSFET	Moselund (2008) Ref. [RL2]	Dutta 2017 (LIF neuron) Ref. [RL44]	-	Singh (2022), Kadam (2024) (simulations only) Refs. [RL45,RL46]	Pending
Charge-trapping Transistor	Kothandaraman (2015) Ref. [RL9]	Gu (2017) (synapse) Ref. [RL10]	-	[Qiao2022] (crossbar vector-matrix multiply) Ref. [RL11]	> 7 years
eFLASH / Floating Gate	Chen (1977) Ref. [RL47]	Diorio (1996) (synapse) Ref. [RL48]	Guo (2017) Ref. [RL49]	[Wang2023] [RL50]	21 years

The reason is that moving from single device to ANN still requires an immense engineering work. In all cases, these large neuromorphic systems employ synaptic devices based on standard memory array architectures, that are adapted from common digital or mixed signal circuits. Moreover, a common aspect of all these neural network implementations is that they require a very large amount of custom-designed, dedicated peripheral circuits, interfaces with external circuitry for testing, and several iteration cycles of circuit, block and system level design, fabrication and test before a functional prototype can be achieved. In fact in multiple cases, neuron activations are either implemented with external hardware or even emulated

numerically with a system in the loop (FPGA or a PC). In fact, we have extensively discussed this in a recent review ^[RL51]. We are convinced that, the fact that we employed standard widely-used MOSFET transistors to implement electronic neurons and electronic synapses (using punch-through impact ionization phenomenon) will result in much shorter development time towards full hardware implementation of ANNs, compared to the other technologies (see Table R1).

Now, asking us to do both things at the same time and report them in the same article seems unfair to us, because nobody has ever achieved such thing using any type of phenomenon/device. Note that many of the articles in Table R1 are publications in Nature, reporting either the use of the phenomenon to mimic neurons/synapses or the construction of an ANN, but never both things in the same article. Hence, we wish the reviewer can relax this request. Furthermore, our manuscript reaches the length limitations imposed by Nature editors, and it has 46 pages of Supplementary information (including 8 Supplementary Notes and 36 Supplementary Figures).

In the revised version of this manuscript, we have included Table R1 in the Supplementary Information (now Supplementary Table 2), and in the Discussion section we have added a few sentences, as follows:

“In the history of microelectronics, neural and synaptic behaviours based on different physical phenomena have been observed in different types of devices/circuits (see Supplementary Note 7 and Supplementary Table 2 therein). In all cases, the time between the first neuron/synapse demonstration and the first ANN demonstration (hardware implementation, not simulation) was at least 7 years, and in some device technologies (such as ferroelectric field effect transistors) it has still not been achieved. The reason is that moving from single device to ANN still requires an immense engineering work, including the development of a very large amount of custom-designed peripheral circuits, interfaces with external circuitry for testing, and several iteration cycles of circuit, block and system level design, fabrication and test before a functional prototype can be achieved, atop the technology integration challenges that different devices may present. However, the adoption of un conventionally-biased MOSFETs for mimicking neuro-synaptic responses can be a fundamental breakthrough towards accelerating the next generation of neuromorphic computers without incurring into major technology alterations. The MOSFET transistor keeps surprising us and now – after this study – it appears to be the perfect building block to implement artificial neural networks.”

As a last point, the authors assign the synaptic behavior to a single Si transistor with unconventional biasing in the abstract. However, strictly seen, this needs to be corrected as at least two transistors are required (an additional one to tune R_B).

We apologize if our initial explanation was unclear.

In our study we demonstrate that one single transistor can be operated in 3 different regimes: neuron, synapse, or neuro-synaptic device depending on the value of the bulk resistance (R_B). This value of R_B could be adjusted during the design of the transistor through contact engineering, for example silicide block on the bulk contact. As our CMOS microchip has not been optimized for any of these three regimes (we use as-provided standard devices), we used a second transistor to adjust R_B , but the second transistor is not needed if R_B can be tuned during the fabrication – which is the case.

The second transistor is only necessary to build a versatile cell capable of operating in any of the three regimes at will, as the second transistor is only used to change the R_B of the first transistor (when applying a gate voltage in the second transistor).

Therefore, the different electronic behaviours obtained in our experimental conditions, can be achieved in different conditions within a 1- or a 2-transistor cell. In detail:

- Generating the electrical output of regime of leaky integrate and fire neuron, such as that depicted in Fig. 1d, and measured in Fig. 2e, can be done on a single transistor with a sufficiently large bulk-contact resistance as to provide an increase in bulk potential due to impact ionization currents. This was observed in single transistors measured across multiple dice of the same wafer in revised Supplementary Fig. 5. Since our devices have not been engineered to operate under these conditions, device-to-device homogeneity of the process can be poor, as highlighted by the differences observed between devices also in Supplementary Fig. 5.
- Generating the electrical output regime of the synapse is achieved by a single transistor, where the bulk terminal is strictly disconnected (effectively floating). This operating conditions facilitate charge-trapping processes and allows the device to display long-term memory effects. We have provided new TCAD simulations and experiments contained in a new Supplementary Note 4 where the conditions behind this regime are discussed in detail.
- Generating the continuously tuneable neuro-synaptic regime does require the inclusion of a second transistor in order to provide full control of the bulk contact connection. In this configuration. In this configuration, the second device connected to the bulk provides this degree of control through its gate voltage V_{G2} .

To clarify this, we have revised the abstract to clearly convey the number of devices that need to be used in each regime. This part of the abstract now reads:

“Here we show that a single transistor can exhibit excellent neural and synaptic behaviours if biased in a specific (unconventional) manner. By connecting one additional CMOS transistor in series, we build a versatile 2-transistors-cell that exhibits adjustable neuro-synaptic response. This electronic performance comes with an impressive yield of 100% and an ultra-low device-to-device variability, owing to the maturity of the silicon CMOS platform used — no materials or devices alien to the CMOS process are required.”

[1] https://theses.hal.science/tel-02506292v1/file/PARK_2019_archivage.pdf (page 34, Figure 2.1)

We deeply thank the reviewer for the insightful reference.

Referee #3 (Remarks to the Author):

With this paper the authors tackle the problem of area over-head of state-of-the-art neuron and synapse circuits by proposing a 2-transistor cell circuit that if biased properly can show: a. neuron and synapse dynamics and b. synaptic plasticity mechanisms. The authors propose to operate the device on the verge of punch-through conditions exploiting the resistance of the bulk connection to ground to switch between neuron, synaptic and plasticity mode. The authors show that this 2-transistor cell can reliably produce multiple neuro-synaptic behaviors.

To the best of my knowledge the papers shows an original approach to realize neuromorphic circuits and it could significantly drive to the field to more efficient and smaller implementations.

We deeply thank the reviewer for taking the time to review our work and for highlighting its originality and potential.

However, it is not clear from the Introduction whether this is the first attempt to use the transistor in this regime for neuromorphic applications or not. If it is, the authors should state it more clearly. If it is not, the authors should cite related references.

We appreciate the reviewer for rising this comment. Yes, this is the first attempt to use the transistor in these regimes for neuromorphic applications. Our study presents the transistor in three regimes: 1) a purely leaky integrate and fire neuron regime (discussed around Fig. 2 in the main text), 2) a purely synaptic regime (discussed around Figs. 4 and 5 in the main text), and 3) a continuously tuneable neuro-synaptic regime which allows controlling neuron dynamics (firing and relaxation times, inhibition) and synaptic plasticity response (discussed around Fig. 3 in the main text).

The operation in regimes 2 and 3 has never been reported before in a standard CMOS single- or 2-transistor device. The operation in regime 1 (neuron) has been investigated before in floating-body standard transistors by one group led by Ganguly et al.^[RL44-RL46,RL52-RL54] This implementation effectively employs the same floating-bulk conditions to mimic only neural dynamics. However, in all cases, neuron behaviour is achieved by inserting the floating-bulk device as integrator element within a much larger circuit comprising, at least, 8 additional transistors^[RL45,RL46] (for firing detection and feedback of a reset signal) or even larger circuits with amplifiers and embedded delays (capacitors and inverter chains) to obtain a specific, fixed neuron response^[RL44]. This results in larger area requirements for each neuron when compared to the 2-transistor cell we propose. Moreover, the flexibility we display, where the simple 2-transistor cell performs a wide range of functions with large configurability, has not been displayed before.

Other implementations of non-standard floating-body semiconductor devices have been proposed in the past for their use in neuron mimicking circuits, as we discussed in a full paragraph of the introduction of our original manuscript (lines 81-101) and within Supplementary Table 1 and Supplementary Note 1 of the original version of our manuscript.

For clarity, in the revised version of the manuscript we have:

- 1- Revised the Introduction section to clearly convey the most representative cases of floating-body devices that target neural and/or synaptic behaviours with standard technologies in the main text. In the revised Introduction, paragraph 3 now reads:

"In this scenario, it is logical to seek alternatives within existent technology platforms (see Supplementary Table 1 and Supplementary Note 2). Two studies have proposed the implementation

of electronic neurons and synapses based on home-made floating-gate silicon transistors^{18,19}, but the performance demonstrated was limited (especially for the synapse); more importantly, fabricating these devices is more complex and expensive than CMOS transistors, and their integration density is lower both due to architecture and to nitridation thermal budgets²⁰. One study implemented electronic neurons using partially depleted silicon-on-insulator (PDSOI) tunnelling²¹ field-effect transistors (FET), but tunnelling FETs require specific doping profiles and gate alignment approaches. So far, the most technology-ready alternative to build electronic neurons is the use of standard PDSOI transistors operating in a band-to-band tunnelling regime²²⁻²⁵ (used as an integrator element in a leaky integrate and fire neuron); however, this approach still requires the interconnection of 6 devices to realize the thresholding operation. More importantly, demonstrations with PDSOI transistors (both tunnelling and standard) can only mimic (limited) neuron functions, not synapses.”

- 2- Discussed in more detail other non-standard technologies to the revised Supplementary Table 1 and revised Supplementary Note 2.

The paper is well written, and the results are clearly explained. Multiple neuron-synaptic behavior has been reported such as threshold voltage neuron modulation, firing capabilities from 20Hz to 200kHz, synaptic potentiation and depression and retention of the synaptic update up to 2 hours. Moreover, a control on the bulk bias of the MOSFET of the 2-transistor cell circuit enables robust and reliable results.

I have no suggestions for improvement on the current content of the paper, but I would like the authors to address my following comment/curiosity.

We are very grateful towards the reviewer for appreciating the depth of the displayed results.

It would be interesting if the authors could extend the last part of the discussion and explain how they would integrate these 2-transistor cell in a large-scale network where neurons are connected to groups of synapses. If the idea is to use the same 2-transistor cell to implement synapses and neurons, how would the synaptic current excite the neuron circuit. How would the current from the neuron circuit be read out and how would the communication between different neurons be handled?

We greatly appreciate this enquiry by the reviewer. The implementation of these devices in a wider system is indeed an interesting discussion point. Due to extension editorial constraints, we have introduced a new Supplementary Note 6 discussing implementation aspects of our proposed device within more elaborate neural networks. Within it, we have included a new Supplementary Fig. 32 (reproduced in this letter as Fig. R15) with a schematic representation displaying a possible array level implementation of a neuromorphic core based on floating bulk devices (which, of course, comprises some peripheral circuitry as in all ANN hardware implementations). The fundamental architecture of the artificial neural network would be very similar to those based on other technologies that rely on non-volatile memory as synaptic devices, such as NOR-flash or memristor technologies.

From the technological point of view, an important aspect should be considered when integrating these devices in large arrays. If neurons are expected to be tuned individually, independent body regions are necessary. SOI technologies are excellent candidates for this, as has been shown in other neuron implementations exploiting floating bulk dynamics within large circuits involving more than 7 transistors, capacitors and other components (see Supplementary Table 1). In bulk silicon technologies, triple wells and deep trench isolation are good alternatives to achieve this, with the corresponding area overhead, but neuron density is less challenging than synaptic array density. In synaptic arrays, many synaptic floating bulk transistors can share a common floating bulk region, but the crossbar addressing circuitry must ensure only

the selected devices are being subjected to the writing voltage conditions and rest of the devices connected to the same drain line should be floated to minimize disturbance (half-select) during each cell write. Similar addressing strategies are already common in non-volatile memory arrays employing other charge-trapping technologies, including NAND/NOR Flash [RL50] and charge trapping transistors [RL11].

Fig. R15 | Schematic representation of a potential neural network section implemented with floating bulk transistor neurons and synapses. The array implementation is similar to floating gate or memristor array-based in-memory compute cores. The floating bulk neuron could be operated with **a**, current input, and monitoring its spiking action potential (see Fig. R16) or **b**, in voltage mode by using the gate as input (see panel **c**), which would require different peripherals to condition the output for the next synaptic array layer.

The array architecture can be very similar to in-memory compute cores based on floating gate or memristive synapses (see Fig. R15). The peripherals would include array addressing decoders, switches and multiplexers to change the array between training and inference conditions, output sense amplifiers that gather the synaptic crossbar signal and feed it to the output neurons. In general, in between layers of the neural network, buffering amplifiers and signal level shifters or amplifiers are required to condition the signal before is fed to the next layer of neurons through a separate synaptic array. The most convenient architecture for connecting the proposed neuron would be to operate it in current input mode. For this end, each column of the synaptic array could be directly connected to the drain of a neuron cell, although current buffers are recommended (to avoid the neuron loading the output of the synaptic columns). Meanwhile, a voltage buffer can sense the voltage at the neuron to obtain output voltage spikes (as will be discussed further along this letter in the bursting mode). Another alternative is to convert the output of the crossbar columns with transimpedance amplifiers and operate the neuron in voltage mode, requiring a current readout at the output to detect the firing condition. Voltage input mode in the floating bulk neuron is basically driving the gate of the device at constant drain voltage, which will depend on the designed threshold required from the neuron within the neural network. There are several strategies to arrange the peripheral circuits in hardware-based artificial neural networks, some of which are discussed in a recent review [RL51] (see an example for a memristive network of 2T2R differential synaptic arrays in Fig. R16).

(continue in next page)

Fig. R16 | Detail of the control circuits used for the dual inference/write procedures. a, Complete circuit schematic for a 4x8 1T1R crossbar array. **b**, Detail of the synchronizers including the sense amplifiers used to detect the correct programming of a given memristor. **c**, Address block, essentially a counter which sequentially addresses each memristor in the crossbar. **d**, Row and column decoders, used to enable the memristor addressed by the address block. **e**, Row and column driver, used to bias the rows with the voltage input or with the programming signal, and to connect the columns to the output neurons (during inference) or the sense amplifier (during write-verify).

Additionally, to experimentally demonstrate some possibilities, we have carried out additional experiments based on our devices. Particularly, in the revised version of our manuscript we provide the detailed characterization of the tuneable neuron operating under bursting mode (current stimuli, voltage output). This configuration directly represents a readout implementation of the neuron in the context of a Spiking Neural Network (SNN), where the neurons receive excitatory signals from the synaptic connections and generate periodic action potentials at its output depending on the input level and the sensitivity of the neuron itself. We elaborate on this experiment below.

Tuneable frequency bursting neuron.

In the original version of the manuscript, we showed that the floating bulk device can produce a configurable integrate and fire neuron behaviour. Integrate and fire functions fundamentally consist of accumulating potential from multiple temporal inputs for a finite time and, when the accumulated potential exceeds a threshold, they produce a temporal output. While the integrative aspect is provided by the

generation and recombination rate of excess carriers in the verge of punch-through impact ionization conditions, the firing dynamics are provided by the parasitic bipolar transistor in the MOSFET structure, that can be finely tuned via a bulk connected control device. This already poses an important advantage compared to other emerging devices used for this end, such as diffusive memristors: while in memristors the response time is pretty much fixed by the materials involved (electrodes and switching medium) with a certain dependence on the input magnitude, the 2-transistor neuron allows for a wide range of response times within a single, simple device.

However, the integrate and fire neuron is a first order behaviour of a neuron in its simplest terms. Individual biological neurons are well known to be capable of higher order functions in specific roles. Particularly, neurons that play fundamental roles in the production of motor, sensory, and cognitive behaviours produce periodic action potentials, or “bursts” at the output once a certain input threshold excitation is reached. In its simplest form, neurons can produce periodic spikes in response to a constant input. If this period (frequency) depends on the input magnitude of the excitatory signal, the process is known as spike number adaptation^[RL55].

Some memristors have shown certain bursting capabilities determined, once again, by the materials involved in its structure^[RL55]. A typical approach is to use simple thresholding memristors within a relaxation oscillator, requiring a couple extra components. Approaches involving CMOS devices include similar neuron implementations that have been recently demonstrated using partially depleted silicon-on-insulator MOSFETs^[RL45,RL46]. However, these devices operate in a band-to-band tunnelling regime, which requires negative gate bias, and also require multiple additional circuits to generate a reset signal through the bulk control transistor in order to operate in bursting mode. Moreover, the spiking frequency can be tuned in the range 1 KHz – 1 MHz, which is much higher than the frequencies of biological processes, typically a drawback in multiple bio-mimicking approaches that require large area and power overheads to reach biological regimes^[RL56].

Fig. R16 | Experimental testbench for floating bulk transistor in tuneable bursting neuron mode. The floating bulk transistor is connected in the two-device configuration to a current source, a voltage amplifier and an oscilloscope.

The experiment testbench (see Fig. R16) is run by contacting an individual 180 nm transistor with the probe station and configured in the two-transistor device using an external off-the-shelf transistor (BS170). The drain of the floating bulk device is connected, via probe tips, to a current source (SMU in current source mode) and to a high-input impedance voltage follower built with off-the-shelf components (TLC2262). The objective is to minimize the load on the output of the floating bulk device. The output of the amplifier drives the input impedance of a passive compensated oscilloscope probe connected to a digital sampling oscilloscope (MSO-X 3024G). All the instruments are controlled by a single MATLAB script that forces a constant excitatory current through the SMU and immediately launches an oscilloscope acquisition. The excitatory current is swept in steps and the firing frequency of the neuron is measured in the oscilloscope and in the MATLAB script based on the captured waveform. V_G and V_{G2} are also parametrically swept to

cover the whole range of tuneable frequencies of the floating bulk device. Typical neuron firing results for different biasing conditions are displayed in Fig. R17, and the tuning range of the firing frequency is displayed in Fig. R18 as a function of input excitatory current. By considering different biasing points for V_G and V_{G2} , the configurable neuron can burst in the range of frequencies 1 Hz ~ 2 KHz, well in agreement with the characteristic firing times reported in Fig. 3. These frequency values are an excellent fit to mimic biological processes of neuro-receptors, such as retinal or cochlear sensory cells^[RL56], and the bursting behaviour of neurons observed from cells in motor and positioning centres of the brain^[RL57], displaying the versatility of the 2-transistor neuron. This is achieved for input excitatory currents between 1 nA/ μm ~ 20 $\mu\text{A}/\mu\text{m}$ (total current is proportional to transistor width), with a roughly linear dependence of frequency on input bias.

Fig. R17 | Oscillatory spiking behaviour of tuneable floating bulk neuron. Using V_G , V_{G2} , and the value of the excitatory current, the floating bulk device can show configurable spiking across a wide range of frequencies. Note that low frequencies (panel a) are damped by parasitic capacitance effects of the characterization setup.

Fig. R18 | Range of firing frequencies across the tuning space and input excitatory currents. The 2-transistor floating bulk neuron device bursts through 3 orders of magnitude in frequency depending on the input current.

Unfortunately, the manuscript is already very long, and Nature imposes rigorous manuscript length based on word count. Hence, we have:

- 1- Inserted one sentence in the revised manuscript, second paragraph of the Discussion section, to briefly introduce this point. This passage now reads:

“An attractive characteristic that highlights from our results is the potential of a very simple and compact device of mimicking multiple neuro-synaptic behaviours. From the neural perspective^{8,28}, in addition to the threshold voltage modulation, the control of the bulk biasing network can be used as refractory period modulation, excitatory or inhibitory signal sensitivity tuning, and spike frequency adaptation. The same building block can operate in current or voltage mode (see Supplementary Figs.29-32), provide fast leaky-integrate-and-fire or spike frequency adaptation over a wide range of bursting frequencies and input stimuli (see Supplementary Note 6), accommodate external reset signals (via bulk bias control pulsing) and operate as a two-, three- or four-terminal neuron within different neural network architectures⁴⁵ (see Supplementary Note 6 and Supplementary Fig.33). From the synaptic perspective, a single device could in principle replace SRAM (a volatile memory cell comprising at least 6 transistors, see Fig.1e) in binarized weight neural networks, or embedded FLASH in multilevel synaptic arrays, with the immediate advantage of a significant area and cost reduction per bit. Moreover, synaptic behaviours here presented can potentially be enhanced in MOSFETs using metal-oxide high-k dielectrics (HfO_x), as these materials allow charge trapping and de-trapping in a much higher degree than SiO₂ (used in this study)⁴⁶.”

- 2- The rest of the discussion, including system level aspects and adaptable frequency bursting has been included as a new Supplementary Note 6 and new Supplementary Figs. 29-33 therein.

References

- RL1. Taur, Y. & Ning, T. H. *Fundamentals of Modern VLSI Devices*. (Cambridge University Press, 2009).
- RL2. Moselund, K. E. *et al.* Punch-through impact ionization MOSFET (PIMOS): From device principle to applications. *Solid-State Electronics* **52**, 1336–1344 (2008).
- RL3. Li, Y., Lee, J.-W. & Sze, S.-M. Optimization of the Anti-Punch-Through Implant for Electrostatic Discharge Protection Circuit Design. *Jpn. J. Appl. Phys.* **42**, 2152 (2003).
- RL4. Guegan, G. *et al.* A 0.10 μm buried p-channel MOSFET with through the gate boron implantation and arsenic tilted pocket. *Solid-State Electronics* **46**, 343–348 (2002).
- RL5. Takeuchi, H. *et al.* Punch-Through Stop Doping Profile Control via Interstitial Trapping by Oxygen-Insertion Silicon Channel. *IEEE Journal of the Electron Devices Society* **6**, 481–486 (2018).
- RL6. Fujiwara, H. *et al.* 34.4 A 3nm, 32.5TOPS/W, 55.0TOPS/ mm^2 and 3.78Mb/ mm^2 Fully-Digital Compute-in-Memory Macro Supporting INT12 \times INT12 with a Parallel-MAC Architecture and Foundry 6T-SRAM Bit Cell. in *2024 IEEE International Solid-State Circuits Conference (ISSCC)* vol. 67 572–574 (2024).
- RL7. Li, C. *et al.* Short-term synaptic plasticity in emerging devices for neuromorphic computing. *iScience* **26**, 106315 (2023).
- RL8. Pazos, S. *et al.* Solution-processed memristors: performance and reliability. *Nat Rev Mater* **9**, 358–373 (2024).
- RL9. Kothandaraman, C. *et al.* Oxygen vacancy traps in Hi-K/Metal gate technologies and their potential for embedded memory applications. in *2015 IEEE International Reliability Physics Symposium MY.2.1-MY.2.4* (2015). doi:10.1109/IRPS.2015.7112816.
- RL10. Gu, X. & Iyer, S. S. Unsupervised Learning Using Charge-Trap Transistors. *IEEE Electron Device Letters* **38**, 1204–1207 (2017).
- RL11. Qiao, S., Moran, S., Srinivas, D., Pamarti, S. & Iyer, S. S. Demonstration of Analog Compute-In-Memory Using the Charge-Trap Transistor in 22 FDX Technology. in *2022 International Electron Devices Meeting (IEDM) 2.5.1-2.5.4* (2022). doi:10.1109/IEDM45625.2022.10019527.
- RL12. AG, I. T. Embedded Flash IP Solutions - Infineon Technologies. <https://www.infineon.com/cms/en/product/memories/embedded-flash-ip-solutions/>.
- RL13. Khan, F., Cartier, E., Woo, J. C. S. & Iyer, S. S. Charge Trap Transistor (CTT): An Embedded Fully Logic-Compatible Multiple-Time Programmable Non-Volatile Memory Element for High-k -Metal-Gate CMOS Technologies. *IEEE Electron Device Letters* **38**, 44–47 (2017).
- RL14. Rauch, S. E. & Guarin, F. The energy driven hot carrier model. in *Hot Carrier Degradation in Semiconductor Devices* 29–56 (Springer International Publishing, 2015). doi:10.1007/978-3-319-08994-2_2.
- RL15. Saks, N. S. *et al.* Observation of hot-hole injection in NMOS transistors using a modified floating-gate technique. *IEEE Transactions on Electron Devices* **33**, 1529–1534 (1986).
- RL16. Yoshikawa, K. *et al.* Lucky-hole injection induced by band-to-band tunneling leakage in stacked gate transistors. in *International Technical Digest on Electron Devices* 577–580 (1990). doi:10.1109/IEDM.1990.237132.
- RL17. Ielmini, D., Ghetti, A., Spinelli, A. S. & Visconti, A. A study of hot-hole injection during programming drain disturb in flash memories. *IEEE Transactions on Electron Devices* **53**, 668–676 (2006).
- RL18. Pazos, S., Aguirre, F., Palumbo, F. & Silveira, F. Reliability-aware design space exploration for fully integrated RF CMOS PA. *IEEE Transactions on Device and Materials Reliability* **20**, 33–41 (2020).
- RL19. Garba-Seybou, T., Federspiel, X., Bravaix, A. & Cacho, F. New Modelling Off-state TDDB for 130nm to 28nm CMOS nodes. in *2022 IEEE International Reliability Physics Symposium (IRPS) 11A.3-1-11A.3-7* (2022). doi:10.1109/IRPS48227.2022.9764431.

- RL20. Song, W. *et al.* Programming memristor arrays with arbitrarily high precision for analog computing. *Science* **383**, 903–910 (2024).
- RL21. Rao, M. *et al.* Thousands of conductance levels in memristors integrated on CMOS. *Nature* **615**, 823–829 (2023).
- RL22. Ovshinsky, S. R. Reversible Electrical Switching Phenomena in Disordered Structures. *Phys. Rev. Lett.* **21**, 1450–1453 (1968).
- RL23. Kuzum, D., Jeyasingh, R. G. D. & Wong, H.-S. P. Energy efficient programming of nanoelectronic synaptic devices for large-scale implementation of associative and temporal sequence learning. in *2011 International Electron Devices Meeting* 30.3.1-30.3.4 (2011). doi:10.1109/IEDM.2011.6131643.
- RL24. Suri, M. *et al.* Phase change memory as synapse for ultra-dense neuromorphic systems: Application to complex visual pattern extraction. in *2011 International Electron Devices Meeting* 4.4.1-4.4.4 (2011). doi:10.1109/IEDM.2011.6131488.
- RL25. Tuma, T., Pantazi, A., Le Gallo, M., Sebastian, A. & Eleftheriou, E. Stochastic phase-change neurons. *Nature Nanotech* **11**, 693–699 (2016).
- RL26. Sebastian, A. *et al.* Temporal correlation detection using computational phase-change memory. *Nat Commun* **8**, 1115 (2017).
- RL27. Ambrogio, S. *et al.* Equivalent-accuracy accelerated neural-network training using analogue memory. *Nature* **558**, 60–67 (2018).
- RL28. Le Gallo, M. *et al.* A 64-core mixed-signal in-memory compute chip based on phase-change memory for deep neural network inference. *Nat Electron* **6**, 680–693 (2023).
- RL29. Hickmott, T. W. Low-Frequency Negative Resistance in Thin Anodic Oxide Films. *Journal of Applied Physics* **33**, 2669–2682 (1962).
- RL30. Snider, G. S. Spike-timing-dependent learning in memristive nanodevices. in *2008 IEEE International Symposium on Nanoscale Architectures* 85–92 (2008). doi:10.1109/NANOARCH.2008.4585796.
- RL31. Prezioso, M. *et al.* Training and operation of an integrated neuromorphic network based on metal-oxide memristors. *Nature* **521**, 61–64 (2015).
- RL32. Moll, J. L. & Tarui, Y. A new solid state memory resistor. *IEEE Transactions on Electron Devices* **10**, 338–338 (1963).
- RL33. Ishiwara, H. I. H. Proposal of Adaptive-Learning Neuron Circuits with Ferroelectric Analog-Memory Weights. *Jpn. J. Appl. Phys.* **32**, 442 (1993).
- RL34. Mulaosmanovic, H. *et al.* Novel ferroelectric FET based synapse for neuromorphic systems. in *2017 Symposium on VLSI Technology* T176–T177 (2017). doi:10.23919/VLSIT.2017.7998165.
- RL35. Jerry, M. *et al.* Ferroelectric FET analog synapse for acceleration of deep neural network training. in *2017 IEEE International Electron Devices Meeting (IEDM)* 6.2.1-6.2.4 (2017). doi:10.1109/IEDM.2017.8268338.
- RL36. Wang, Z. *et al.* Experimental Demonstration of Ferroelectric Spiking Neurons for Unsupervised Clustering. in *2018 IEEE International Electron Devices Meeting (IEDM)* 13.3.1-13.3.4 (2018). doi:10.1109/IEDM.2018.8614586.
- RL37. Kim, I.-J., Kim, M.-K. & Lee, J.-S. Highly-scaled and fully-integrated 3-dimensional ferroelectric transistor array for hardware implementation of neural networks. *Nat Commun* **14**, 504 (2023).
- RL38. Soliman, T. *et al.* First demonstration of in-memory computing crossbar using multi-level Cell FeFET. *Nat Commun* **14**, 6348 (2023).
- RL39. Berger, L. Exchange interaction between ferromagnetic domain wall and electric current in very thin metallic films. *Journal of Applied Physics* **55**, 1954–1956 (1984).
- RL40. Vincent, A. F. *et al.* Spin-Transfer Torque Magnetic Memory as a Stochastic Memristive Synapse for Neuromorphic Systems. *IEEE Transactions on Biomedical Circuits and Systems* **9**, 166–174 (2015).
- RL41. Jung, S. *et al.* A crossbar array of magnetoresistive memory devices for in-memory computing. *Nature* **601**, 211–216 (2022).

- RL42. Song, M. Y. *et al.* High RA Dual-MTJ SOT-MRAM devices for High Speed (10ns) Compute-in-Memory Applications. in *2023 International Electron Devices Meeting (IEDM)* 1–4 (2023). doi:10.1109/IEDM45741.2023.10413832.
- RL43. Han, J.-K., Yu, J.-M. & Choi, Y.-K. A Junctionless Single Transistor Neuron With Vertically Stacked Multiple Nanowires for Highly Scalable Neuromorphic Hardware. *IEEE Transactions on Electron Devices* **69**, 3142–3146 (2022).
- RL44. Dutta, S., Kumar, V., Shukla, A., Mohapatra, N. R. & Ganguly, U. Leaky Integrate and Fire Neuron by Charge-Discharge Dynamics in Floating-Body MOSFET. *Sci Rep* **7**, 8257 (2017).
- RL45. Singh, A. K., Saraswat, V., Baghini, M. S. & Ganguly, U. Quantum Tunneling Based Ultra-Compact and Energy Efficient Spiking Neuron Enables Hardware SNN. *IEEE Transactions on Circuits and Systems I: Regular Papers* **69**, 3212–3224 (2022).
- RL46. Kadam, A. A., Singh, A. K., Somappa, L., Baghini, M. S. & Ganguly, U. A Compact Low Power Multi-mode Spiking Neuron using Band to Band Tunneling. in *2024 IEEE International Symposium on Circuits and Systems (ISCAS)* 1–5 (2024). doi:10.1109/ISCAS58744.2024.10557876.
- RL47. Chen, P. C. Y. Threshold-alterable Si-gate MOS devices. *IEEE Transactions on Electron Devices* **24**, 584–586 (1977).
- RL48. Diorio, C., Hasler, P., Minch, A. & Mead, C. A. A single-transistor silicon synapse. *IEEE Transactions on Electron Devices* **43**, 1972–1980 (1996).
- RL49. Guo, X. *et al.* Fast, energy-efficient, robust, and reproducible mixed-signal neuromorphic classifier based on embedded NOR flash memory technology. in *2017 IEEE International Electron Devices Meeting (IEDM)* 6.5.1-6.5.4 (2017). doi:10.1109/IEDM.2017.8268341.
- RL50. Wang, G. *et al.* A 40nm 5-16Tops/W@INT8 eFlash In-Memory Computing SoC Chip with Noise Suppression and Compensation Techniques to Improve the Accuracy. in *2023 IEEE International Conference on Integrated Circuits, Technologies and Applications (ICTA)* 128–129 (2023). doi:10.1109/ICTA60488.2023.10363786.
- RL51. Aguirre, F. *et al.* Hardware implementation of memristor-based artificial neural networks. *Nat Commun* **15**, 1974 (2024).
- RL52. Dutta, S., Bhattacharya, T., Mohapatra, N. R., Suri, M. & Ganguly, U. Transient Variability in SOI-Based LIF Neuron and Impact on Unsupervised Learning. *IEEE Transactions on Electron Devices* **65**, 5137–5144 (2018).
- RL53. Dutta, S., Chavan, T., Mohapatra, N. R. & Ganguly, U. Electrical Tunability of Partially Depleted Silicon on Insulator (PD-SOI) Neuron. *Solid-State Electronics* **160**, 107623 (2019).
- RL54. Chavan, T., Dutta, S., Mohapatra, N. R. & Ganguly, U. Band-to-Band Tunneling Based Ultra-Energy-Efficient Silicon Neuron. *IEEE Transactions on Electron Devices* **67**, 2614–2620 (2020).
- RL55. Kumar, S., Wang, X., Strachan, J. P., Yang, Y. & Lu, W. D. Dynamical memristors for higher-complexity neuromorphic computing. *Nat Rev Mater* **7**, 575–591 (2022).
- RL56. Milozzi, A., Ricci, S. & Ielmini, D. Memristive tonotopic mapping with volatile resistive switching memory devices. *Nat Commun* **15**, 2812 (2024).
- RL57. Fox, D. M., Rotstein, H. G. & Nadim, F. Bursting in Neurons and Small Networks. in *Encyclopedia of Computational Neuroscience* (eds. Jaeger, D. & Jung, R.) 1–17 (Springer, New York, NY, 2013). doi:10.1007/978-1-4614-7320-6_454-1.

Referee #1 (Remarks to the Author):

A. Summary of key results

The authors demonstrate that a single conventional MOSFETs can be used to implement neurons and (tunable) synapsis. This would certainly be a great advantage for the implementation of neuromorphic circuits in conventional CMOS technology. The experimental results are very clear and complete, covering the full range of bio-mimicking devices. In fact, using a transistor connected to the bulk terminal, the authors show that it is possible to modulate the punch-through avalanche mechanism in the transistor to show either an abrupt thresholding behavior (neurons), a configurable time-dependent synaptic plasticity behavior or a synaptic behavior.

B. Originality and significance

The paper is fully novel to the best of my knowledge. The author's proposal seems to be disruptive for the implementation of neurons and synapses using a single conventional MOS transistor. The fact that the authors deal with a conventional "deterministic" device based on well-known physics is certainly a strong advantage over other less conventional proposals. In this regard, the single weak point is that the long-term synaptic behavior is controlled by charge trapping in the gate insulator. In any case, my recommendation is to publish the paper because it is really novel and significant.

C. Data and methodology: validity of the approach, quality of data, quality of presentation.

The authors present a wide variety of data, covering all fundamental aspects of the proposal. In my opinion, their approach is adequate, the presented data is what is required for the purpose of the paper and the results are clearly presented. As in the previous manuscript, I don't see any issue with the experimental methodology.

I also must point out that, as I suggested in my previous review, the authors have complemented their experimental data with TCAD simulations (and not only with a compact model). I think that this has substantially improved the quality of the paper.

D. Appropriate use of statistics and treatment of uncertainties.

The required statistics are sufficiently reported.

E. Conclusions: robustness, validity, reliability

The experimental results cannot be debated and constitute the main piece of contribution of the paper. The interpretation is also solid though sometimes it is not easy to explain all the details through the analysis of the device basic operation, and elements such as traps in the insulator or parasitic capacitances are invoked. I am not raising a criticism here, I am only saying that the involved physics are complicated. In summary, my opinion is that the author's conclusions are robust, valid and reliable.

F. Suggested improvements: experiments, data for possible revision

In my previous revision I made several suggestions and the authors have been able fully fulfilled my expectations. In particular, the most important ones are:

- I raised the question of long-term retention of several states in the synaptic regime that should include some kind of trapping in the device. The authors have described the process in terms of trapping in the oxide.

- I suggested some measurements with different ramp rates in the neuron regime. The authors have made these measurements and provided an understandable discussion of the results.

- I suggested to complement the experimental results with TCAD simulations and they have included these results in this second version of the manuscript.

As far as I can see, I also consider that the authors have adequately replied to the other reviewers reports.

I have no further comments or suggestions and I recommend publication.

G. References: appropriate credit to previous work?

To the best of my knowledge, the references are fully appropriate.

H. Clarity and context: lucidity of abstract/summary, appropriateness of the abstract, introduction and conclusions.

The manuscript is very clear and nicely contextualized with respect to the state-of-the-art. Abstract, introduction and conclusions are also OK for me. In fact, after revision of the first manuscript, the conclusions are stronger and more reliable.

We would like to deeply thank the reviewer for taking the time to read our manuscript and for the valuable feedback provided. The depth of the provided feedback by the reviewer was invaluable for strengthening the manuscript and we sincerely appreciate the dedication towards the peer review process. We are happy all concerns have been satisfactorily answered.

Referee #2 (Remarks to the Author):

The reviewer would like to thank the authors for their extensive answers to my concerns and updating their manuscript accordingly. There are no more concerns from my side.

We would like to deeply thank the reviewer for taking the time to read our manuscript and for the valuable feedback provided. We are happy all concerns have been satisfactory answered.

Referee #3 (Remarks to the Author):

I would like to thank the authors for addressing my questions. I have no further comments.

We would like to deeply thank the reviewer for taking the time to read our manuscript and for the valuable feedback provided.

This is really a very complete, novel and interesting paper. Being able to use single conventional MOSFETs as neurons and (tunable) synapses would certainly be a great advantage for the implementation of neuromorphic circuits. The experimental results are very clear and complete, covering the full range of bio-mimicking devices. In fact, using a transistor connected to the bulk terminal, the authors show that it is possible to modulate the punch-through avalanche mechanism in the transistor to show either an abrupt thresholding behavior (neurons), a configurable time-dependent synaptic plasticity behavior or a synaptic behavior. The fact that the authors deal with a conventional “deterministic” device based on well-known physics is certainly a strong advantage over other less conventional proposals.

In general, I agree with the authors discussion but, there is an important point that I am not able to fully understand and which generates me some significant doubts (at least, as far as the interpretation is concerned). My main doubt is related to the behavior of the devices as synapses. The authors show that their devices can show long-term and short-term plasticity (LTP and STP). In this regard, I can qualitatively understand the STP behavior (although the relaxation times are so long that the underlying physics are also not fully clear to me). On the contrary, in the extreme case of fully floating bulk ($V_{G2}=0$), **I have serious doubts about the physical interpretation of the LTP behavior in these MOSFETs.**

Main comment/objection

My main objection is related to the role of charges accumulated in the bulk of the transistor, particularly when they behave as synapses.

I can understand that during punch-through/avalanche there is a buildup of charge in the body (and in the channel) that changes the potential distribution in the device. This can certainly explain the volatile behavior observed when the devices behave as neurons. This is observed when the backgate transistor resistance is small and favors carrier collection. In this case, when the drain reaches a threshold voltage (which depends on the ramp rate), punch-through and impact ionization are triggered and the current suddenly increases. When V_{DS} is reduced after this punch-through/snapback transition, the potential distribution has changed and lower values of V_{DS} can sustain impact ionization and hence the accumulation of charges. This accumulation of charges disappears when V_{DS} reaches the reset voltage. This explains why there is a memory window. It is also true that in this case, the bulk connection favors the collection of carriers. In this case, I can understand the role of accumulated charges (mainly holes) because all the effects are observed under high drain bias and there is a continuous carrier supply (from impact ionization in the drain).

Although I can qualitatively understand the role of charges when the devices are operated as neurons, there is one detail that I would also like to clarify for this operation mode. If the V_{DS} backward sweep of (Suppl. Fig. 4a/b and main text page

3 line 163), the reset voltage is supposedly related to recombination of carriers in the parasitic bipolar transistor base (collection from the bulk as well). This is explained in supplementary note 1 page 4, lines 124-130 and in other places of the manuscript. However, no dependence on the sweep rate is observed during the backward sweep. This seems to be in contradiction with the fact that the transition is related to the relaxation of excess of carriers and the return to “equilibrium”, one would expect that faster sweeps would lead to smaller reset voltages because the carrier needs a certain time to relax. **Can the authors clarify this point?**

In the case of “floating body” conditions, when the devices behave as synapses, I cannot understand how these carriers remain in the device for such long times (as shown in Fig. 4c and Suppl. Fig 15). Notice that the effects remain even at low drain bias, when impact ionization cannot occur so that no extra carriers are supplied. It is true that the floating body connection severely limits the collection of the carriers but there is still recombination, and recombination times are indeed much shorter than 2 hours. And there are also leakage currents through all the terminals of the device, i.e. holes can flow to the source and drain terminals since positive charges in the body forward biases these junctions. It is very difficult for me to understand how a cloud of carriers can remain in the device even when transistor bias is removed ($V_{DS}=0$). **How can a large population of holes remain in the device for such a long time (at least two hours) unless they are trapped somewhere?** If we look at other devices proposed as synapses, when they are programmed by appropriate electrical signals, there is always a long lasting internal modification leading to a nearly stable state of a certain resistance. In the case of floating gate transistors it is the trapped charge in the floating gate, in RRAM it is the formation/destruction of a ionic filament (either metal atoms or oxygen vacancies), in FeFETs the ferroelectric polarization, etc. It is very difficult for me to believe that in the MOSFETs considered in this work this internal variable is the presence/absence of a cloud of “permanent” carriers in the bulk. **This is my main objection and I would like the authors to explain this in detail.**

In the introduction the authors state that *“The connections between multiple processing units are 57 known as synapses, which need to change their electrical resistance in a non-volatile manner to favor (facilitate) or limit (depress) the connections between specific neurons”*. Thus, a defining property of a synapse is non-volatily”. They claim that their device can behave as synapses but afterwards they say (with respect to different stable states when under floating body conditions: *“Though this process should not be interpreted as non-volatility (from its memory conception), this synaptic state retention characteristic is certainly long-term, as the weight is maintained through more than 2 hours without requiring any type of dynamic update or refresh”*. I don’t understand the difference between “non-volatility” and long-term retention. I think that the authors should clarify this point. Why they say that their states are non-volatile but behave as synapses? Is it a question of the retention time? Why the authors mention this subject even if they don’t have any indication of memory loss during their experiments (Fig. 4c)?

Although I am not questioning the experimental results, but the physics-based interpretation of the results is extremely important. If the device behavior is due to an intrinsic effect of MOSFETs biased in an unconventional way (punch-through/impact ionization), then, it would be feasible to properly design the devices with desired properties. If, on the contrary, the effects are related to trapping of charges somewhere in the device, this would be a serious drawback.

The authors present results from a compact model. However, assessing the physics of these devices under the reported conditions would require a physics-based simulation. There are several commercial packages which allow to do these simulations in a relatively easy way. **I strongly encourage the authors to do this type of physics-based simulations.** In particular, this might allow to unveil the microscopic nature of the experimentally observed behaviors and to respond to my main objection related to synaptic behavior and charge accumulation.

Other comments and suggestions

- I suggest that, in the interest of the general reader it would be convenient to introduce a supplementary note explaining the details of what is the difference between (a) integrate-and-fire neural behavior, (b) tunable neuro-synaptic plasticity and (c) synaptic behavior.
- In the case of memristive devices, the origin of cycle-to-cycle (C2C) variability can be understood at least qualitatively (changes in the filament in RRAM, changes in the melted volume in PCRAM, etc.). However, it is harder for me to understand C2C variability in a well-control device such as a MOSFET based on deterministic processes (unless charge trapping were involved). What is the physics-based origin of the cycle-to-cycle variability (both the ON and OFF resistances) as shown in Fig. 3a and suppl. Figs. 9 and 10? Can the authors comment on this?
- The conditions under which retention measurements are performed (Fig. 4c, suppl. Fig. 15) should be specified (V_G, V_{DS}). What would happen if V_{DS} were kept at zero volts and retention measured by successive pulses? Would the device relax to “equilibrium” during the $V_{DS}=0$ periods? I suggest to do some measurements in this regard.

Some minor corrections

- Page 5, line 266: *we operate 500 nm thick-oxide transistors in the fully floating-body condition (V_{G2}) with a bipolar voltage scheme.* Shouldn't it be $V_{G2}=0$?
- The description of Fig5(b) seems incomplete ¿what is the difference between the left and the right curves (resistance versus CDF)? By the way, more

standard plots would show CDF versus the independent variable (R in this case) and not vice versa.

- The explanation of the sweep rate dependence of the “threshold voltage (V_{DS})” seems reasonable in the main text page 3, line 160-162. However, the explanation in the figure caption of Suppl. Fig 4a/b is in contradiction with it.
- The pulse amplitude for depression and the reading voltage are different in Fig 4(b) and in the description of the same figure in the main texts (page 6 line 285).
- The values of potentiation and depression voltages in Suppl. Fig. 16 do not coincide with those reported in the text (Page 6 line 304)
- In Suppl. Fig 11, the bulk gate voltage is named “ V_{Gb} ”. In the text, it is V_{G2} . This is the same in Fig. 3d and 3e (in the caption it is ok)